**Resource**

EMBO
**Molecular Medicine**

# Protein-based tools for the detection and characterisation of Oropouche virus infection

Monique K Merchant [1], Juliano de Paula Souza[2], Sahar Abdelkarim[3,4], Shrestha Chakraborty [1],
Tufi A Nasser Neto [2], Kristel I Gutierrez Manchay[2], Daniel M de Melo Jorge[2], Valdinete Alves do
Nascimento[5], Yuxian Sun [1], Eve R Caroe[1], Lauren E A Eyssen [3], Jared S Rudd[2], Isa C Ribeiro Piauilino[6,7],
Sérgio Damasceno Pinto [6], Matheus H Pereira da Silva [2], Felipe Rocha do Nascimento[2],
Felipe Gomes Naveca [5,8], José L Proenca-Modena [9], Luis L P da Silva[2], Regina M Pinto de Figueiredo[6],
Raymond J Owens[3,4], Eurico Arruda [2✉] & Stephen C Graham [1✉]

## Abstract

**Oropouche fever is a neglected tropical disease caused by the orthobunyavirus Oropouche virus (OROV). A recent OROV epidemic caused by a novel reassortant has seen infections across an expanded geographical range, with deaths of healthy adults plus vertical transmission leading to pregnancy loss. OROV research and epidemiology is hampered by a paucity of available tools for serology and molecular virology. We have purified recombinant OROV nucleoprotein and the spike region of the viral surface glycoprotein Gc. These antigens detect seroconversion following experimental infection of animals in indirect ELISA, confirming their antigenic authenticity. They stimulate the production of high neutralising antibody titres in animals, highlighting their promise as immunogens for vaccination. We developed a nanobody-based sandwich ELISA that can detect OROV antigens in human clinical serum samples with high efficiency, and we show that nanobodies directed against OROV Gc spike can potently neutralise infection by both historical OROV strains and the newly emerged reassortant. Our protein-based reagents will accelerate OROV research and highlight the utility of protein-based tools for future OROV vaccines and point-of-care diagnostic devices.**

**Keywords** Neglected Tropical Disease (NTD); Brazil; Amazon; Arbovirus; Vector-borne Disease
**Subject Categories** Methods & Resources; Microbiology, Virology & Host Pathogen Interaction; Proteomics

## Introduction

Oropouche virus (*Orthobunyavirus oropoucheense*; OROV) is the cause of Oropouche fever, an arthropod-borne disease endemic to Latin America and the Caribbean (Travassos da Rosa et al, 2017; Sakkas et al, 2018). First identified in Trinidad and Tobago in 1955 (Anderson et al, 1961), OROV causes acute illness characterised by fever, headache, myalgia and arthralgia (Mourão et al, 2009), and it can cross the blood-brain barrier to cause aseptic meningoencephalitis (Bastos et al, 2012; Vernal et al, 2019). Despite causing numerous outbreaks in the Amazon basin over the last 65 years (reviewed in (Wesselmann et al, 2024; Files et al, 2022)), OROV remains a neglected tropical disease. The similarity of symptoms of OROV infection to those caused by co-circulating arboviruses like Dengue and Chikungunya viruses prevents the diagnosis based purely on clinical presentation and mean that the prevalence of OROV infection is likely to be underestimated (Sakkas et al, 2018; Durango-Chavez et al, 2022; Bastos et al, 2012).

Since late 2023, there has been a dramatic increase in the number of OROV cases detected across Brazil, Bolivia, Colombia, Peru and Cuba (World Health Organization, 2024; Gräf et al, 2025). OROV has a tri-segmented RNA genome and this increase in cases is likely to have been driven by the emergence of a novel reassortant (Scachetti et al, 2025; Naveca et al, 2024). While the primary clinical presentation of infection by this new epidemic reassortant is similar to that seen in previous outbreaks (Naveca et al, 2024), there have been several reports of vertical transmission in Brazil that may have caused foetal death, miscarriage or microcephaly of the newborn (Martins-Filho et al, 2024; Garcia Filho et al, 2024). This outbreak has also seen the first two fatal OROV infections in adults, with two women in their twenties from a nonendemic region of Brazil succumbing to the infection

[1]Department of Pathology, University of Cambridge, Cambridge CB2 1QP, UK. [2]Department of Cell Biology and Center for Virus Research, Ribeirão Preto Medical School, University of São Paulo, Ribeirão Preto, São Paulo 14049-900, Brazil. [3]The Rosalind Franklin Institute, Harwell Science Campus, Didcot OX11 0QX, UK. [4]Division of Structural Biology, University of Oxford, The Wellcome Centre for Human Genetics, Oxford OX3 7NB, UK. [5]Núcleo de Vigilância de Vírus Emergentes, Reemergentes ou Negligenciados – ViVER/EDTA, Instituto Leônidas e Maria Deane, Fiocruz, Manaus, Amazonas 69053-138, Brazil. [6]Fundação de Medicina Tropical Dr. Heitor Vieira Dourado, Manaus, Amazonas 69040-000, Brazil. [7]Programa de Pós-graduação em Medicina Tropical, Universidade do Estado do Amazonas (UEA), Manaus, Amazonas 69050-010, Brazil. [8]Laboratório de Arbovírus e Vírus Hemorrágicos, Instituto Oswaldo Cruz, Fiocruz, Rio de Janeiro, Rio de Janeiro 21040-900, Brazil. [9]Laboratory of Emerging Viruses, Department of Genetics, Evolution, Microbiology and Immunology, Institute of Biology, University of Campinas, Campinas, São Paulo 13083-862, Brazil. ✉E-mail: eaneto@fmrp.usp.br; scg34@cam.ac.uk

(Bandeira et al, 2024). Notably, in both fatal cases the clinical course was remarkably similar to that of severe Dengue virus infection, emphasising the need for definitive molecular diagnosis to correctly ascribe severe febrile illnesses aetiology.

Diagnosis of OROV infection can be achieved via RT-PCR analysis of serum samples collected within 5–7 days of symptom onset (Weidmann et al, 2003; Wise et al, 2020; Pan American Health Organization (PAHO), 2024), and RT-PCR analysis can be multiplexed to also detect co-circulating viruses like Mayaro virus (Naveca et al, 2017). However, RT-PCR analysis is often unavailable in primary points-of-care in epidemic regions, and OROV infection is not always considered, leading to significant underdiagnosis (Alva-Urcia et al, 2017; Martins-Luna et al, 2020). Multiple groups have developed in-house serological tests for OROV infection, including ELISA and immunofluorescence assays (summarised in (Wesselmann et al, 2024)), but these have only recently become commercially available. A recent study raised monoclonal antibodies that could detect OROV infection in experimentally infected mice (immunohistochemistry) or cultured cells (immunocytochemistry), but its utility for detecting acute OROV infection in humans was not determined (Andreolla et al, 2024).

The dearth of detection reagents for OROV antigens impedes the development of point-of-care diagnostics tests like lateral flow devices and slows the pace of OROV molecular virology research. In this study, we generated a panel of Variable Heavy-chain domains of camelid Heavy-chain antibodies (VHHs; a.k.a. nanobodies) (Harmsen and De Haard, 2007) that specifically detect the nucleoprotein and surface glycoprotein of OROV. Bivalent formulations of these nanobodies can detect OROV antigens in the serum of patients with acute OROV infection and potently neutralise OROV infection in vitro. These detection reagents open new avenues for the development of low-cost protein-based detection assays to detect and antiviral therapies to treat OROV infection. Furthermore, we show that the antigens used to generate the nanobodies elicit strong neutralising antibody titres in mice, highlighting their potential for use as vaccine immunogens.

## Results

Orthobunyaviruses form enveloped particles that enclose an RNA genome tightly bound by the nucleoprotein (N) and possess the glycoproteins Gn and Gc on their surface (Fig. 1A) (Hughes et al, 2020). The N-terminal region of Gc forms a prominent 'spike' on the surface of orthobunyavirus particles, composed of conserved 'head' and 'stalk' domains (Bowden et al, 2013; Hellert et al, 2019; Hover et al, 2023). Previous studies have shown that OROV infection elicits antibodies directed against the proteins N and Gc (Saeed et al, 2001; Barbosa et al, 2023), suggesting that both these antigens are accessible to the humoral immune system. OROV N was thus purified following recombinant expression in E. coli, and the Gc spike domain (residues 482–894) was purified following expression using mammalian (Freestyle 293F) cells to allow protein glycosylation (Fig. 1B). Purified OROV N had a 260:280 nm absorbance ratio ≥1.5, consistent with purification of N bound to bacterial RNA (Murillo et al, 2018). Purified antigens were used as capture antigens in an indirect ELISA and incubated with serum

from mice experimentally infected with OROV. Both N and Gc spike are recognised by IgG (Fig. 1C) and IgM (Fig. EV1A) from serum of infected mice. Neither antigen is recognised by antibodies present in a mock-infected mouse, nor is there recognition of the Gc spike from the related Cristoli orthobunyavirus (CRIV; Figs. 1C and EV1A). This confirms that the purified recombinant antigens display authentic epitopes that are presented to the immune system during OROV infection. To test their ability to raise a neutralising immune response, purified OROV antigens (Gc spike with or without N) were used to immunise 6-week-old mice using a prime-boost-boost regimen. Both antigens raised a strong, specific IgG response (Fig. EV1B) that was potently able to neutralise OROV infection in vitro (Fig. 1D).

Purified OROV N and Gc spike were used to immunise a llama, and nanobodies were isolated, enriched and cloned following the procedure outlined in (Eyssen et al, 2024). Sequence analysis following enrichment with phage display identified multiple distinct clusters for each antigen and, for each antigen, six nanobody sequences that were representative of each unique cluster (Fig. EV2A; Table EV1) were selected for further characterisation. Nanobodies were cloned and purified following E. coli expression with a His$_8$ tag alone or with a His$_9$ tag plus biotin acceptor peptide (AviTag) to facilitate in vitro biotinylation (Fig. EV2B–D), and indirect ELISA experiments confirmed that the biotinylated nanobodies could recognise purified antigen (Fig. 2A). Sandwich ELISA experiments with purified antigens, performed using passive adsorption of the His$_8$-tagged capture nanobody and detection of the biotinylated nanobody via horseradish peroxidase (HRP)-conjugated streptavidin, identified two competition groups for the Gc spike nanobodies, with one group composed of nanobodies SpB6, SpC6 and SpE8 while the second comprised only SpC7 (Fig. 2B). We did not observe any competition for epitopes in the N protein sandwich ELISA (Fig. 2B). On the assumption that the lack of competition observed for N was a result of its propensity to polymerise when purified following E. coli expression (Murillo et al, 2018), competition for epitopes on N was mapped using competitive biolayer interferometry (BLI). In brief, streptavidin biosensors were loaded with biotinylated nanobodies then incubated with either free N or with N in the presence of 10- to 25-fold molar excess of the competitor nanobody, association of N (or N plus non-competitive nanobody) to the biotinylated nanobody being recorded as an increase in BLI signal. Competitive BLI identified three distinct competition groups: NpA2, NpE2 and NpE3; NpF2 and NpG1; and NpD2 (Figs. 2C and EV2E).

Further sandwich ELISA characterisation identified SpB6:(bio-tinylated)bSpC7 and NpF2:bNpE3 as the optimal pairings for capture:detection of Gc spike and N, respectively, able to detect picogram quantities of purified OROV antigens (Fig. 2D). These nanobodies recognise laboratory-grown stocks of OROV strains BeAn19991 and TRVL, with the N nanobodies demonstrating higher sensitivity (Fig. 2E). However, preliminary experiments demonstrated that sandwich ELISAs using these nanobodies were insufficiently sensitive to detect OROV antigens in serum samples from human patients with acute OROV infection.

Multiple parallel improvements were attempted to enhance sandwich ELISA sensitivity. The nanobodies were cloned as fusions with the linker region plus Fc domains 2 and 3 of human IgG1, thereby generating bivalent detection reagents resembling heavy-chain-only antibodies. The resultant dimerisation, driven by the

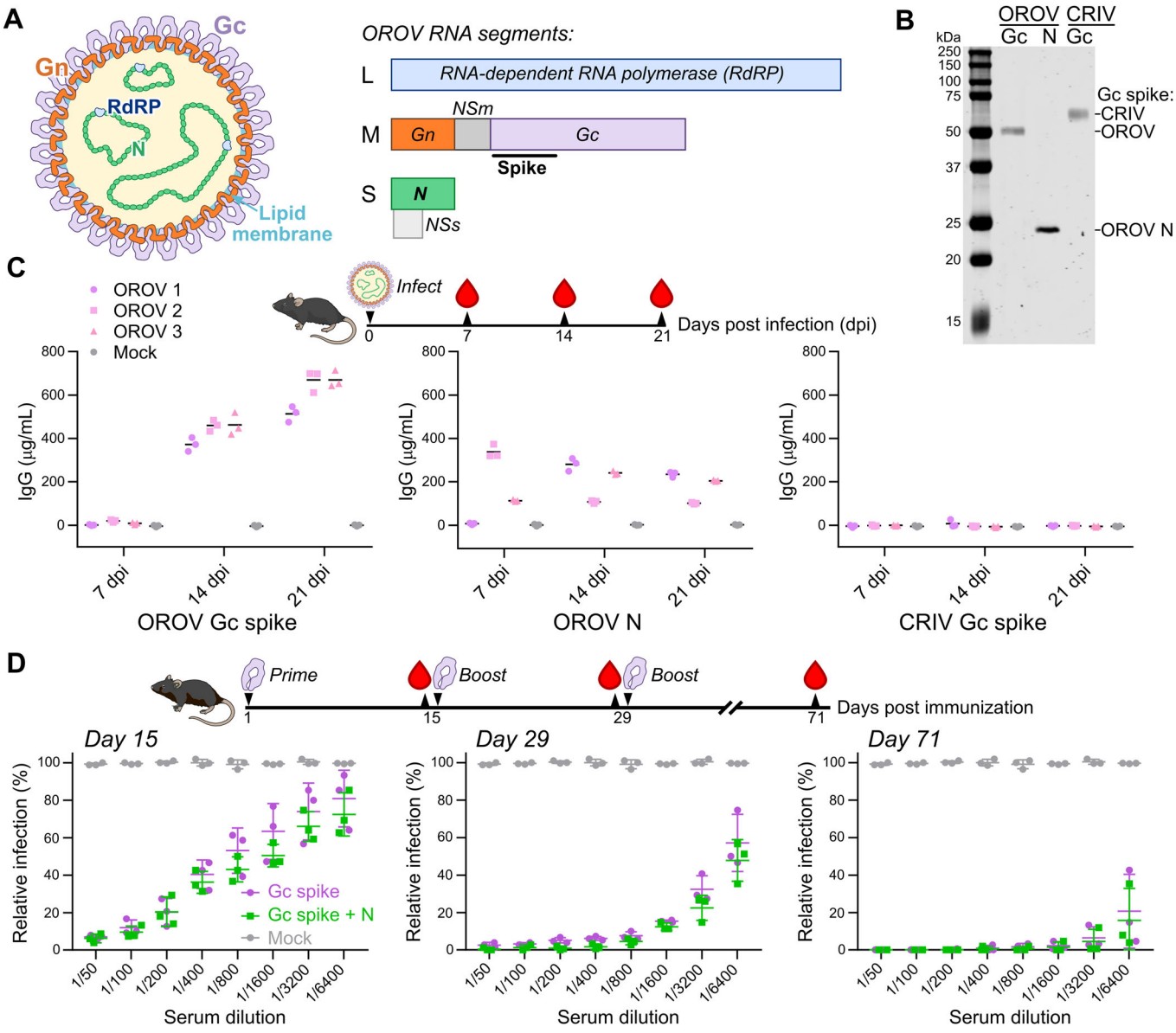

**Figure 1. Purification and characterisation of recombinant OROV antigens.**

(A) Schematic representation of OROV virion (left) and RNA genome segments (right). The RNA-dependent RNA polymerase (RdRP) associates with the circular genome segments and is encoded by the large (L) segment. The surface glycoproteins Gn and Gc, plus the non-structural protein NSm, are encoded by the medium (M) segment polyprotein. The genome-associated nucleoprotein (N) and non-structural protein NSs are encoded by the small (S) segment. Nanobodies were raised against the N and the spike region of Gc (residues 482–894 of the M polyprotein). (B) Coomassie-stained SDS-PAGE of purified OROV antigens, and the Gc spike region of the related orthobunyavirus Cristoli virus (CRIV). (C) Purified antigens are recognised by antibodies raised by mice in response to infection with OROV strain BeAn19991. Blood was harvested at indicated days post-infection (dpi), and an indirect ELISA to detect IgG was performed using indicated antigens. Values are shown for three independent ELISA experiments using serum from three infected mice (OROV 1–3) and one mock-infected control mouse. (D) Neutralising antibodies are elicited by immunisation with purified OROV antigens. Mice were immunised with OROV Gc spike alone, or Gc spike plus N, and blood samples were taken at the indicated times. Serum was diluted for plaque reduction neutralisation assays. The percentage infection was calculated as the number of plaques observed for each sample, divided by the mean number observed for the mock-immunised mice. Data for three mice are shown, plus mean and SD. Source data are available online for this figure.

IgG1 domains, was anticipated to confer avidity-enhanced antigen binding (Girt et al, 2021; Richard et al, 2013). In addition to untagged constructs, these nanobody-Fc fusions were cloned with a C-terminal AviTag to facilitate co-translational biotinylation, allowing directional capture on ELISA plates coated with neutravidin that should enhance exposure of the antigen-binding motifs. Additionally, the detection nanobody-Fc fusions were

covalently coupled to the HRP enzyme. For the sake of expediency, only one nanobody-Fc fusion was generated for the N protein (FcNpD2), while two nanobody-Fc fusions were generated Gc spike-directed nanobodies (FcSpC7 and FcSpB6; Fig. EV3). Of these, the bFcNpD2:FcNpD2-HRP combination proved the most sensitive in detecting purified OROV antigens in a sandwich ELISA (Fig. 3A) with a limit of detection (LOD) of $11.3 \pm 7.4$ pg/mL

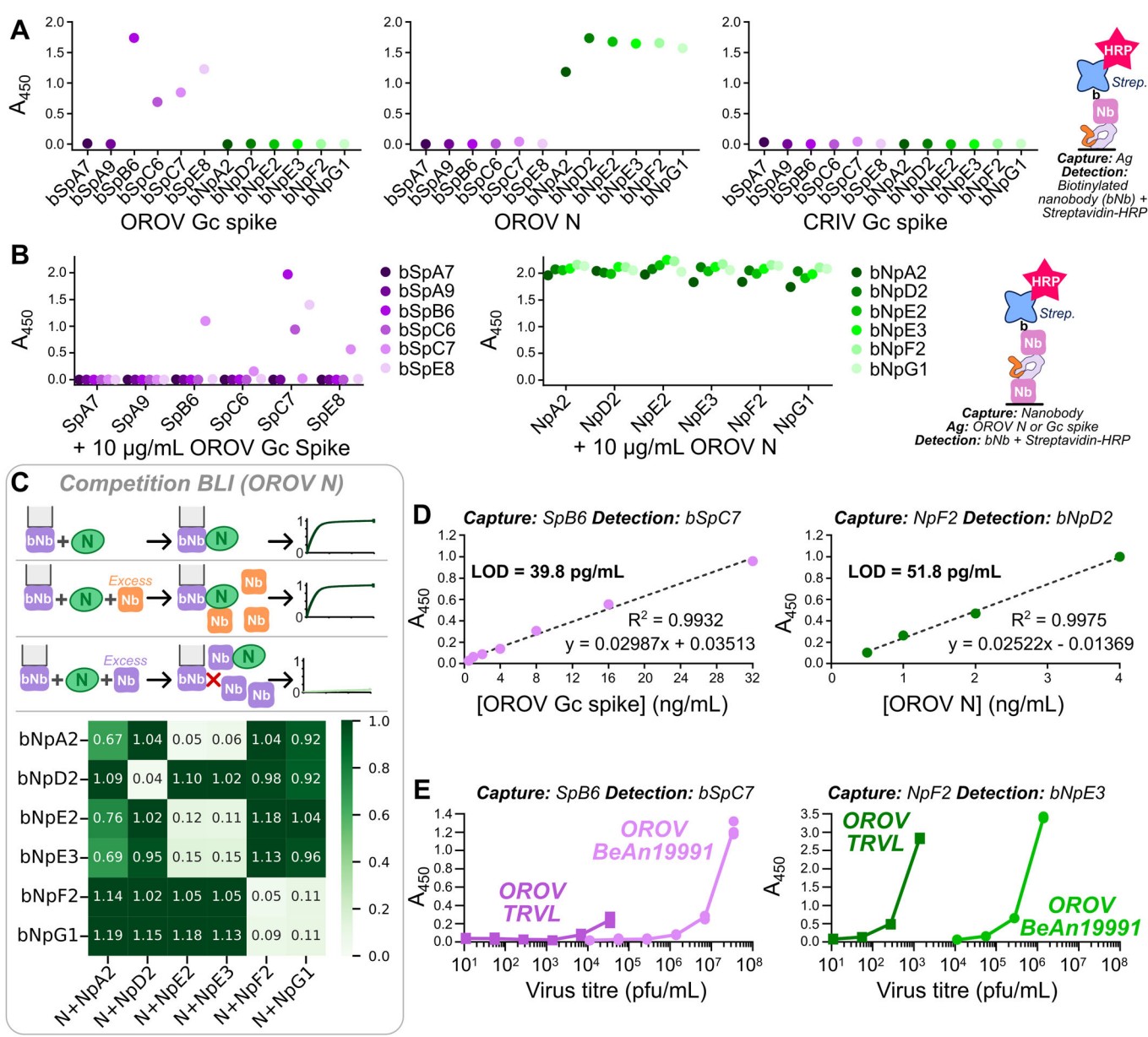

**Figure 2. Nanobodies for specific detection of OROV Gc spike and N.**

(A) Nanobody detection of purified OROV antigens. Antigens were immobilised on the capture surface via passive adsorption (2 µg/mL) and detected via incubation with specified biotinylated nanobodies (bNbs) plus streptavidin-HRP. CRIV Gc spike was included as a negative control. The data represent one experiment. (B) Sandwich ELISA using purified OROV antigens to determine nanobody competition groups. Surfaces were coated with nanobodies (2 µg/mL) before being incubated with antigen then detection bNbs and streptavidin-HRP. For non-competing nanobodies, data are representative of two independent experiments. For competing Gc spike nanobodies the experiment was performed once. (C) Competition biolayer interferometry (BLI) to map nanobody competition for OROV N epitopes. Streptavidin biosensors were loaded with bNbs then incubated with 1 µM OROV N plus 25 µM competitor nanobody. Heatmap shows BLI responses after 180 s of association, normalised relative to the response obtained for each bNb-loaded biosensor incubated with 1 µM OROV N alone. BLI traces are shown in Fig. EV2E. Data are representative of two independent experiments. (D) Limit of detection (LOD) of purified Gc spike (left) or N (right) in sandwich ELISA using optimised nanobody pairings. LOD is calculated as (3.3 * standard deviation [background])/slope. Data are representative of two independent experiments. (E) Detection of Gc spike (left) and N (right) in laboratory stocks of OROV strains TRVL (Anderson et al, 1961) and BeAn19991 (Acrani et al, 2015; Aquino et al, 2003) by sandwich ELISA. Data shown are from one experiment performed in technical triplicate. Source data are available online for this figure.

(mean ± SD, $n = 5$ independent experiments), while the bFcSpC7:FcSpB6-HRP combination had a LOD of 572.1 ± 166.5 pg/mL ($n = 3$ independent experiments). Sandwich ELISAs using the N nanobody-Fc fusions detect laboratory-grown stocks of OROV (Fig. 3B). Importantly, it also recognises the

2023–4 epidemic isolate (AM0088) (Scachetti et al, 2025) of OROV (Fig. 3B). OROV AM0088 N shares 100% amino acid identity with the purified N used for nanobody generation.

The optimised bFcNpD2:FcNpD2-HRP capture:detection sandwich ELISA was used to detect OROV antigens in serum samples

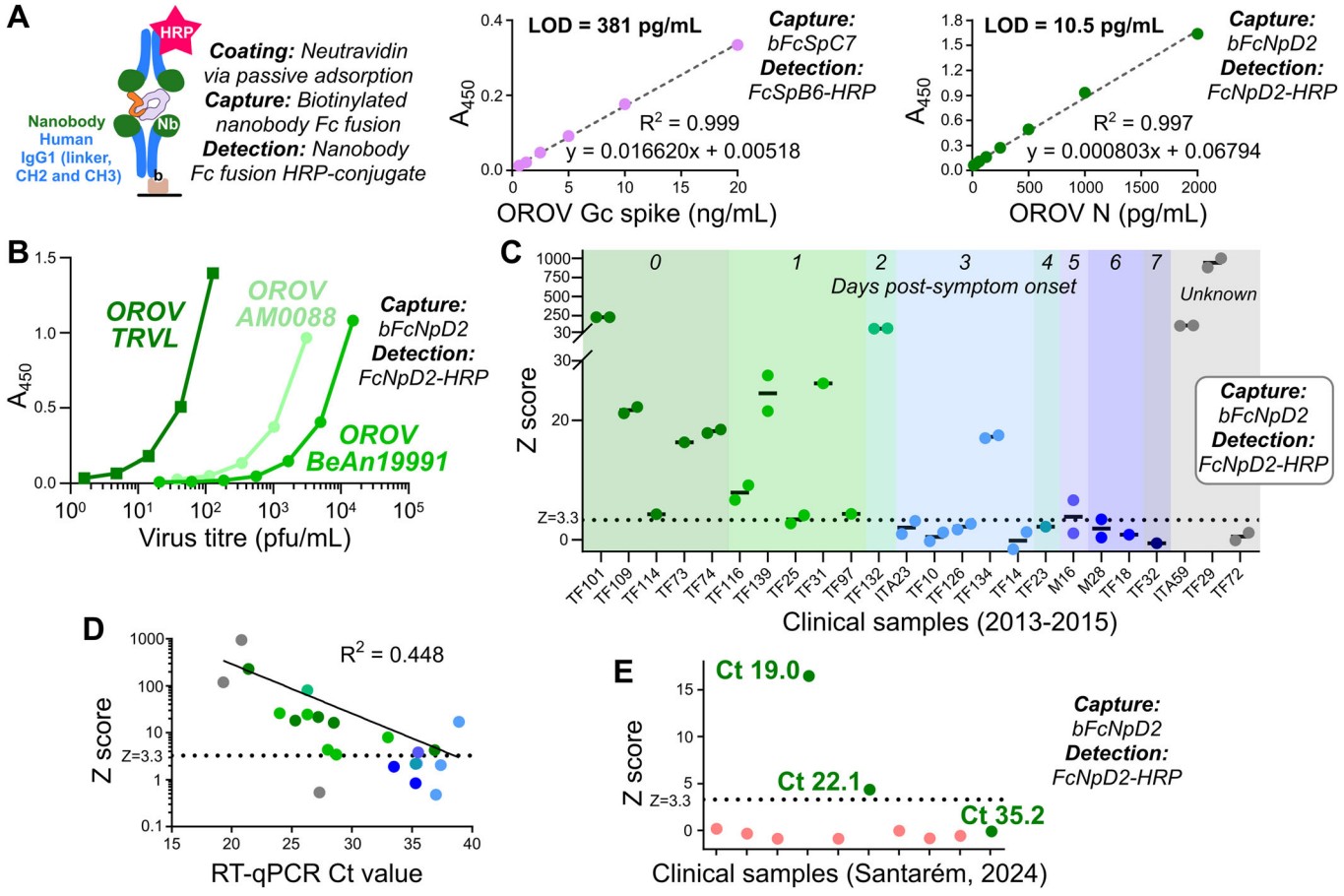

**Figure 3. An improved sandwich ELISA allows detection of OROV N in clinical samples.**

(A) Schematic representation of modified sandwich ELISA (left) plus standard curves to determine LOD for purified OROV Gc spike (middle) and N (right). Data representative of 3 (Gc Spike) or 5 (N) independent experiments. (B) Detection of N in laboratory stocks of OROV strains TRVL (Anderson et al, 1961), BeAn19991 (Acrani et al, 2015; Aquino et al, 2003) and the newly emerged reassortant AM0088 (Scachetti et al, 2025) by sandwich ELISA. (C) Detection of N in a 1:3 or higher dilution of serum from patients infected with OROV between 2013 and 2015 (Table 1). Antigen values are shown as standard deviations above the mean for blank samples (Z score). PCR-confirmed OROV positive samples are grouped by day of collection post symptom onset. Data represent 1 (marked with *) or 2 independent measurements for each sample. (D) Correlation of Z scores from OROV N sandwich ELISA versus RT-qPCR evaluation of viral genome abundance (Ct values). Non-linear regression analysis of the Ct values and $\log_{10}$-tranformed Z scores is shown. (E) Detection of N in a 1:3 dilution of serum from patients with acute febrile illness of unknown aetiology in 2024 in the municipality of Santarém. Subsequent RT-qPCR analysis detected OROV in three samples, for which Ct values are shown (green). Other samples were RT-qPCR negative for OROV (pink). Source data are available online for this figure.

that were collected from patients with acute febrile illness between 2013 and 2015 and confirmed as positive for OROV by RT-qPCR (Table 1). OROV N protein was successfully detected above the noise level (Z score ≥3.3) for all replicates experiments in 13 of 24 samples (Fig. 3C), with a further 3 samples being alternatively above and below the detection limit in duplicate experiments. The level of signal in the sandwich ELISA was correlated to the Ct values obtained by RT-qPCR ($R^2 = 0.448$; Fig. 3D), with robust detection in the sandwich ELISA for samples with Ct values lower than 26. To further test the utility of the sandwich ELISA in the diagnosis of OROV infection, it was used to detect OROV antigens in serum samples from patients with acute febrile illness of unknown aetiology, collected in the city of Santarém in 2024 during the OROV epidemic from ten individuals (8 female, 2 male) aged 23–64 years old. Two of ten samples yielded a signal for OROV above the noise level (Fig. 3E). Subsequent RT-qPCR analysis confirmed both as 'true positives'. There was one sample below the

detection threshold in the ELISA that was RT-qPCR positive, albeit with a relatively modest viral load (Ct = 35.2), and none of the seven samples that were RT-qPCR negative for OROV gave a positive signal in the ELISA.

In addition to their utility for antigen-based diagnostic assays, nanobody-based reagents are highly suitable for use as research reagents in several formats owing to their ease of expression and purification. Immunoblotting of lysates from OROV-infected cells demonstrate that N is recognised by its cognate HRP-conjugated nanobody-Fc fusion while Gc spike is not (Fig. 4A), suggesting that NpD2 recognises a linear epitope while SpB6 and SpC7 recognise conformational epitopes. The N and Gc spike nanobodies were purified from *E. coli* following expression with a C-terminal Cys-His$_8$ tag, allowing covalent conjugation of fluorescent dyes via maleimide chemistry. Fluorescence microscopy showed that both the N and Gc spike nanobodies can detect OROV antigens in HeLa cells experimentally infected with OROV BeAn19991, with

**Table 1. Clinical samples (2013–2015) used in this study.**

| ID | Collection year | Region | Days post symptom onset | Sex | Ct | Z | [N] (pg/mL) |
|---|---|---|---|---|---|---|---|
| M16 | 2013 | Manacapuru | 5 | Female | 35.5 | 3.8 | 51 |
| M28 | 2013 | Manacapuru | 6 | Female | 33.5 | 1.9 | – |
| ITA23 | 2015 | Itacoatiara | 3 | Female | 37.4 | 2.1 | – |
| ITA59 | 2015 | Itacoatiara | Unknown | Male | 19.3 | 118.0 | 1337 |
| TF10 | 2015 | Tefé | 3 | Female | 37.0 | 0.5 | – |
| TF14 | 2015 | Tefé | 3 | Female | 35.3 | −0.2 | – |
| TF18 | 2015 | Tefé | 6 | Female | 35.3 | 0.8 | – |
| TF23 | 2015 | Tefé | 4 | Male | 35.3 | 2.2 | – |
| TF25 | 2015 | Tefé | 1 | Male | 28.7 | 3.4 | – |
| TF29 | 2015 | Tefé | Unknown | Female | 20.8 | 644.1 | 130,180 |
| TF31 | 2015 | Tefé | 1 | Male | 24.0 | 16.7 | 182 |
| TF32 | 2015 | Tefé | 7 | Male | 36.9 | -0.6 | – |
| TF72 | 2015 | Tefé | Unknown | Female | 27.3 | 0.5 | – |
| TF73 | 2015 | Tefé | 0 | Female | 28.5 | 16.3 | 118 |
| TF74 | 2015 | Tefé | 0 | Male | 25.3 | 18.2 | 159 |
| TF97 | 2013 | Tefé | 1 | Male | 28.0 | 4.3 | – |
| TF101 | 2015 | Tefé | 0 | Male | 21.4 | 230.3 | 2663 |
| TF109 | 2015 | Tefé | 0 | Male | 27.2 | 21.7 | 201 |
| TF114 | 2015 | Tefé | 0 | Female | 36.9 | 4.3 | – |
| TF116 | 2015 | Tefé | 1 | Female | 33.0 | 7.9 | 39 |
| TF126 | 2015 | Tefé | 3 | Male | 35.4 | 2.2 | – |
| TF132 | 2015 | Tefé | 2 | Male | 26.3 | 80.2 | 891 |
| TF134 | 2015 | Tefé | 3 | Female | 38.9 | 17.3 | 149 |
| TF139 | 2015 | Tefé | 1 | Female | 26.3 | 24.6 | 237 |

Patients were aged 8–65. All were confirmed as positive for OROV by RT-qPCR. Where available, collection time (days post symptom onset) is show. Ct values from RT-qPCR are shown. Average Z score from sandwich ELISA is shown, as is average concentration of N protein where samples were positive (Z ≥ 3.3) and within the range of the antigen standard curve.

negligible non-specific signal in mock-infected cells (Fig. 4B). The nanobody NpE3 staining substantially overlaps with the signal observed using polyclonal serum obtained from experimentally infected mice, with signal localising primarily to cytoplasmic puncta that may represent virus genome assembly compartments and/or sites of N protein aggregation. There was also signal for nanobody NpE3 in the nucleus of infected cells. For Gc spike, there is limited overlap between nanobody and polyclonal mouse serum signal; nanobody SpE8 signal extensively co-localises with the trans-Golgi marker TGN46, a known site of virus assembly (Barbosa et al, 2018). The signal afforded by polyclonal mouse immune serum is poorer in cells infected with OROV AM0088 (Fig. EV4). The overlap in signal between nanobody SpE8 and TGN46 is less pronounced in these cells, with SpE8 signal at large puncta in the periphery of the cells, and NpE3 staining is again observed at cytoplasmic puncta and in the nucleus (Fig. EV4).

Infection of HeLa cells with OROV BeAn19991 following incubation with nanobody-Fc fusions demonstrates that FcSpB6 and FcSpC7 neutralise infection with 50% neutralising dose ($ND_{50}$) values of 4.9 nM ± 0.79 and 2.1 ± 0.18 nM ($ND_{50}$ ± standard error, $n = 3$ independent titrations), respectively (Fig. 4C). FcSpB6 has substantially reduced ability to neutralise infection by the new

epidemic AM0088 isolate ($ND_{50} = 171 ± 26.3$ nM), whereas FcSpC7 retains potent neutralisation ability ($ND_{50} = 6.4 ± 0.54$ nM).

# Discussion

Here we present a sandwich ELISA that allows detection of the N protein in the serum of patients suffering from acute OROV infection, both from historical outbreaks and from the most recent epidemic. While multiple groups have published IgM capture or indirect ELISAs to detect OROV antibodies (Saeed et al, 2001; Watts et al, 2022; Vasconcelos et al, 2009; das Neves Martins et al, 2025), to the best of our knowledge this is the first demonstration of a sandwich ELISA to detect OROV antigens in clinical samples.

The efficiency of detection was much higher in samples collected within the first 3 days following symptom onset (Fig. 3C). While the signal in the sandwich ELISA is correlated with viral genome abundance as measured by RT-qPCR (Fig. 3D), this assay is currently less sensitive. Sensitivity of the ELISA assay could be improved by using two distinct anti-N protein nanobody-Fc fusions for capture and detection, or by engineering these reagents to encode multiple nanobodies fused to the Fc region to enhance the

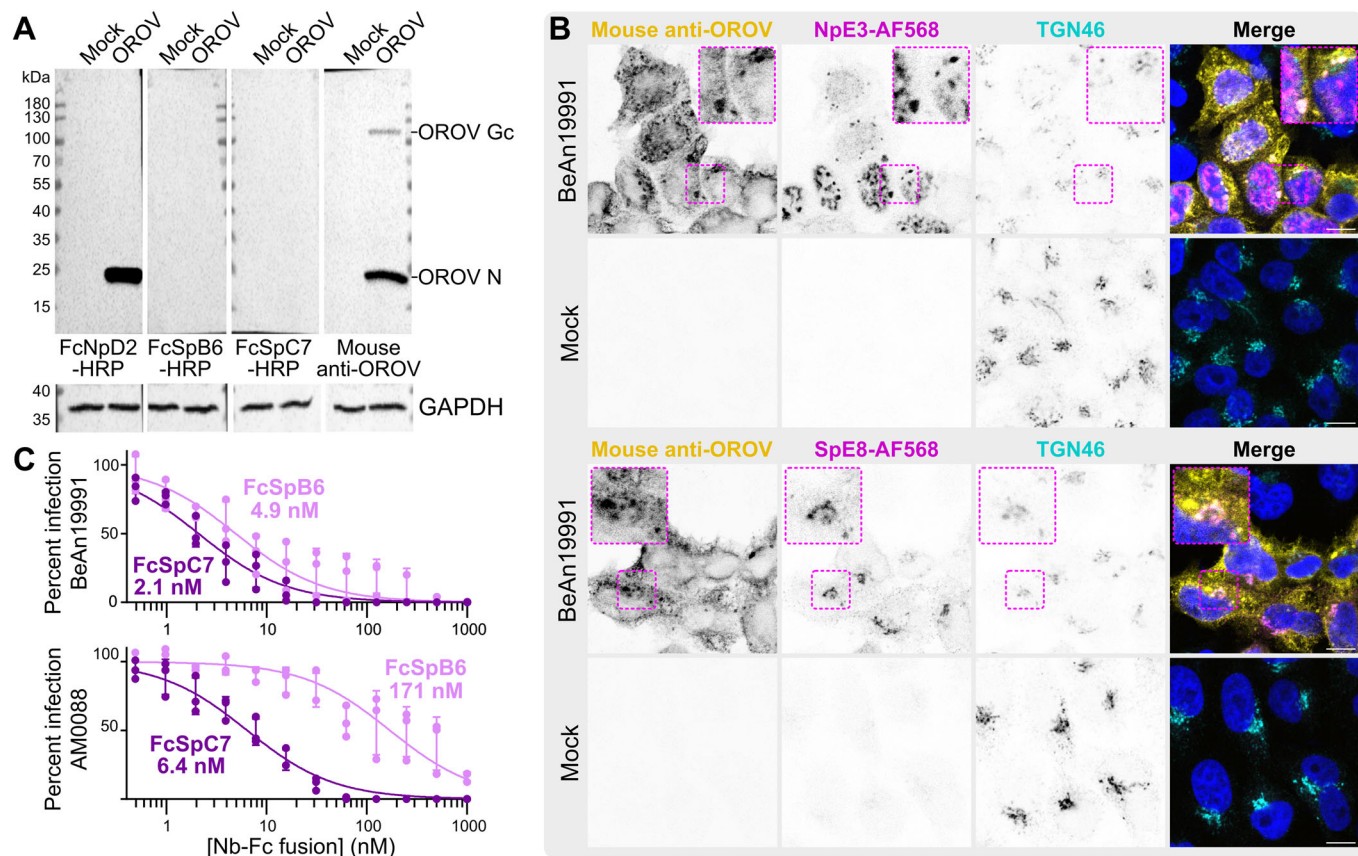

**Figure 4. New tools for OROV research.**

(A) Immunoblots of lysates from HeLa cells mock-infected or infected with OROV BeAn19991 (MOI 5). Membranes were probed with HRP-conjugated nanobody-Fc fusions or with a polyclonal antibody against OROV plus anti-mouse secondary antibody, as indicated. Membranes were subsequently probed for GAPDH as a loading control. (B) Immunocytochemistry of HeLa cells infected with OROV BeAn19991 (MOI 0.5). Cells were probed with AlexaFluor (AF)568 conjugated OROV nanobodies against N (NpE3) or Gc Spike (SpE8), with a polyclonal antibody against OROV and with an antibody against TGN46, plus appropriate secondary antibodies. Nuclei are stained with DAPI (blue). Scale bar = 10 μm. (C) Nanobody-Fc fusions that recognise Gc spike neutralise OROV infection in vitro. Plaque reduction neutralisation assay using Gc spike Nb-Fc fusions, performed using OROV BeAn19991 (top) or AM0088 (bottom). Data for three independent experiments plus SD and $ND_{50}$ values are shown. Source data are available online for this figure.

avidity of binding. We did not observe false positive signals in the seven samples we tested that were negative for OROV by RT-qPCR (Fig. 3E), although further testing is required to define the sensitivity, specificity and predictive values. Having established proof-of-principle that antigen detection can be used for OROV molecular diagnosis, it should be possible to develop a nanobody-based lateral flow device for diagnosing OROV infection at primary points-of-care. Translation from an ELISA to lateral flow format may further reduce the sensitivity of antigen detection, although this could potentially be ameliorated by use of different pairs of capture and detection nanobody-Fc fusions and/or further protein engineering to enhance avidity, as mentioned above. Assuming that adequate sensitivity can be achieved then rapid point-of-care diagnosis of OROV would enhance surveillance efforts, which could be especially important for pregnant women in endemic areas, and it would dramatically accelerate enrolment of patients into trials aimed at determining post-infection sequelae or the efficacy of future therapies.

We have not investigated the cross-reactivity of our nanobodies with other Simbu serogroup N proteins. Of most relevance are the

other Simbu serogroup viruses that were identified in humans, OROV strains Iquitos and Madre de Dios (Ladner et al, 2014; Aguilar et al, 2011), or non-human primates, OROV strain Perdões that was identified in black-tufted marmosets (Tilston-Lunel et al, 2015) and Manzanilla virus first identified in the red howler monkey (Anderson et al, 1960). N proteins from OROV strains Iquitos, Madre de Dios and Perdões share 100% amino acid identity with OROV BeAn19991, thus all would be recognised by the nanobodies presented here. In a clinical context, definitive assignment of the OROV strain infecting an individual would require an orthogonal molecular technique like RT-PCR directed at the M segment (56.6–63.3% nucleotide identity across Oropouche virus strains) and/or full virus genome sequencing (Naveca et al, 2024). Manzanilla virus N (Ladner et al, 2014) shares only 71.2% amino acid identity with OROV N, making cross-reactivity possible but much less likely. Identification of which (if any) OROV N nanobodies detect Manzanilla virus N awaits future experimental confirmation.

The purified antigen used for immunisation of llamas to produce nanobodies was also tested for its ability to elicit

neutralising antibody responses in mice. Immunisation with either OROV Gc spike alone, or Gc spike plus N, raised a strong neutralising antibody response with a 1/6400 dilution of serum taken 6 weeks after the final boost retaining the ability to reduce infection in vitro by ~80% (Fig. 1D). This compares favourably to the only other published immunisation trial for OROV, which used a vesicular stomatitis virus pseudotype system expressing the full OROV M segment or a variant with the Gc head domain removed (Stubbs et al, 2021). Immunisation with the VSV pseudotypes protected mice from clinical symptoms of OROV infection but not from viraemia (Stubbs et al, 2021). Previous studies of Schmallenberg virus (SBV), a related ruminant orthobunyavirus, have shown that immunisation using the Gc head domain or Gc spike (head +stalk) is sufficient to protect mice against lethal challenge (Hellert et al, 2019; Wernike et al, 2017). Furthermore, the majority of neutralising antibodies raised by SBV infection recognise the head domain of Gc (Hellert et al, 2019). Recombinant OROV Gc spike is thus a promising candidate for further vaccine studies using protein or mRNA-based immunisation.

Fc fusions of nanobodies that recognise Gc spike are capable of neutralising OROV infection (Fig. 4C), with FcSpC6 potently neutralising both BeAn19991 and the newly emerged AM0088 reassortant ($ND_{50}$ = 2.1 and 6.4 nM, respectively). Neutralising monoclonal antibodies that recognise the SBV Gc head less potently ($ND_{50}$ = 0.15–1.8 μM) are sufficient to protect mice from lethal challenge (Hellert et al, 2019). This suggests that our Gc spike nanobody-Fc fusions could be used as direct-acting antiviral compounds to prevent or treat OROV infection, although confirmation of this requires further study.

The nanobodies raised against OROV Gc spike comprise two competition groups (Fig. 2B), consistent with previous reports that the spike region of OROV Gc is monomeric in solution (Hellert et al, 2019). The cell surface receptor Lrp1 has recently been identified as mediating OROV infection, likely via binding to the OROV Gn protein (Schwarz et al, 2022). The surface glycoproteins of orthobunyavirus particles undergo dramatic structural rearrangement during viral endocytosis, which may prime the particles for cell entry (Hover et al, 2023). It will be interesting in the future to define the molecular basis of neutralisation by our Gc spike nanobodies, to determine whether they prevent infection by inhibiting a required conformational change, via steric hindrance of Lrp1 binding, or by preventing binding of an additional as-yet unidentified co-receptor.

Our recombinant N protein was contaminated with nucleic acid, as evidenced by the high 260:280 nm absorbance ratio. This is consistent with previous reports that OROV N co-purifies with bacterial RNA when expressed in *E. coli* (Murillo et al, 2018). OROV N forms polymers when encapsidating viral genomes in vivo. Given that OROV N will be bound to RNA in virus particles and infected cells, we considered RNA-bound N to be representative of the antigen likely to be found in patient serum and encountered by the humoral immune system. We observed that nanobodies directed at N do not compete for binding in sandwich ELISA (Fig. 2B), presumably because of N oligomerisation (Murillo et al, 2018). However, competition BLI confirmed that nanobodies raised against OROV N comprise three competition groups (Fig. 2C). For some combinations of biotinylated nanobody, OROV N and competitor nanobody, we observe higher BLI signals than for the biotinylated nanobody with N alone; we ascribe this to a complex of N plus the non-competitive nanobody binding the biosensor, which will be larger and have higher mass than N alone and will thus yield a larger BLI signal.

In immunocytochemistry, there was moderate overlap between the signal from an immune serum raised in mice and anti-N nanobody NpE3 in OROV-infected cells, but we observed more extensive nuclear staining with nanobody NpE3 (Figs. 4B and EV4). This could reflect different epitopes recognised by the immune serum versus NpE3, or the ability of nanobodies to better access epitopes in crowded environments owing to their small size (Loreau et al, 2023). A recent study characterised two novel monoclonal antibodies (mAbs) raised against OROV N by immunisation of mice with OROV-infected-cell supernatant (Andreolla et al, 2024). They demonstrated the utility of their OROV mAbs for immunocytochemistry following experimental infection of several different cell types, and for immunohistochemistry in experimentally infected animals, but they did not report characterisation of the N protein subcellular localisation.

In contrast to N, the overlap between mouse immune serum and anti-Gc spike nanobody SpE8 staining is less extensive in cells infected with OROV BeAn19991 (Fig. 4B). This could reflect the binding of SpE8 to an epitope that is poorly immunogenic in the context of infection or that is inaccessible to antibodies in the immune serum. The glycoproteins of Bunyamwera virus, a related orthobunyavirus, are known to undergo maturation steps at the Golgi upon release that alters their recognition by monoclonal antibodies (Novoa et al, 2005). SpE8 and TGN46 staining overlap extensively in OROV BeAn19991-infected cells, but to a lesser degree in cells infected with OROV AM0088 where punctate SpE8 staining at the cell periphery is more pronounced (Fig. EV4). This suggests potential differences in virus replication or structure between OROV BeAn19991 and AM0088, although we note that the Gc spike sequences of these isolates differ by nine amino acids (Scachetti et al, 2025), and this could affect the efficiency of detection afforded by SpE8.

In summary, we have developed protein-based reagents for the detection and characterisation of OROV infection. Purified OROV antigens enable indirect ELISA serology and show promise as candidate vaccine molecules. Nanobodies that recognise OROV Gc spike and N are valuable research reagents. Fc fusions of nanobodies against N enable detection of OROV antigens in serum samples, opening the way to development of low-cost antigen detection tests, and nanobody-Fc fusions that recognise Gc spike potently neutralise infection by both historical and recently emerged isolates of OROV. The reagents presented here will accelerate laboratory and clinical research into OROV, an important emerging pathogen.

# Methods

**Reagents and tools table**

| Reagent/resource | Reference or source | Identifier or catalogue number |
|---|---|---|
| **Experimental models** | | |
| Freestyle 293F cells | Thermo Fisher Scientific | Cat# R79007 |
| HeLa cells | ATCC | Cat# CCL-2 |
| Vero cells | ATCC | Cat# CCL-81 |

| Reagent/resource | Reference or source | Identifier or catalogue number |
|---|---|---|
| *Escherichia coli* T7 Express lysY/I$^a$ cells | New England Biolabs | Cat# C3013I |
| *Escherichia coli* WK6 cells | ATCC | Cat# 47078 |
| Oropouche virus strain BeAn19991 | Aquino et al (2003) https://doi.org/10.1007/s00705-002-0913-4 | |
| Oropouche virus strain TRVL | Anderson et al (1961) https://doi.org/10.4269/ajtmh.1961.10.574 | |
| Oropouche virus strain AM0088 | Scachetti et al (2025) https://doi.org/10.1016/S1473-3099(24)00619-4 | |
| C57BL/6 mice | Jackson Laboratory | Cat# 000664 |
| **Recombinant DNA** | | |
| OROV N pOPTH | This study | |
| OROV Gc Spike H6 | This study | |
| OROV Gc Spike-BAP-H6 | This study | |
| pMWCAVI_H6BAP-OROV_N | This study | |
| CRIV Gc Spike H6 | This study | |
| pF5A_PBase | This study | |
| PB-RN | Li et al (2013) https://doi.org/10.1073/pnas.1218620110 | |
| OROV Np Nb A2 BAP H9 | This study | |
| OROV Np Nb D2 BAP H9 | This study | |
| OROV Np Nb E2 BAP H9 | This study | |
| OROV Np Nb E3 BAP H9 | This study | |
| OROV Np Nb F2 BAP H9 | This study | |
| OROV Np Nb G1 BAP H9 | This study | |
| OROV Gc Spike Nb A7 BAP H9 | This study | |
| OROV Gc Spike Nb A9 BAP H9 | This study | |
| OROV Gc Spike Nb B6 BAP H9 | This study | |
| OROV Gc Spike Nb C6 BAP H9 | This study | |
| OROV Gc Spike Nb C7 BAP H9 | This study | |
| OROV Gc Spike Nb E8 BAP H9 | This study | |
| OROV Np Nb A2 H8 | This study | |
| OROV Np Nb D2 H8 | This study | |
| OROV Np Nb E2 H8 | This study | |
| OROV Np Nb E3 H8 | This study | |
| OROV Np Nb F2 H8 | This study | |
| OROV Np Nb G1 H8 | This study | |
| OROV Gc Spike Nb A7 H8 | This study | |
| OROV Gc Spike Nb A9 H8 | This study | |

| Reagent/resource | Reference or source | Identifier or catalogue number |
|---|---|---|
| OROV Gc Spike Nb B6 H8 | This study | |
| OROV Gc Spike Nb C6 H8 | This study | |
| OROV Gc Spike Nb C7 H8 | This study | |
| OROV Gc Spike Nb E8 H8 | This study | |
| OROV Np Nb E3 CH8 | This study | |
| OROV Sp Nb E8 CH8 | This study | |
| OROV FcSpB6 | This study | |
| OROV FcSpC7 BAP | This study | |
| OROV FcNpD2 | This study | |
| OROV FcNpD2 BAP | This study | |
| **Antibodies** | | |
| Purified mouse IgG | Merck | Cat# I5381 |
| Purified mouse IgM | Thermo Fisher Scientific | Cat# MA1-10438 |
| HRP-conjugated polyclonal goat anti-mouse IgG | Thermo Fisher Scientific | Cat# 31430 |
| HRP-conjugated polyclonal goat anti-mouse IgM | Merck | Cat# AP128P |
| Anti-GAPDH polyclonal rabbit | Sigma-Aldrich | Cat# G9545 |
| Anti-TGN46 polyclonal sheep | Bio-Rad | Cat# AHP500 |
| Mouse polyclonal antiserum against OROV strain BeAn19991 | Concha et al (2024) https://doi.org/10.1371/journal.ppat.1007047 | |
| HRP-conjugated polyclonal sheep anti-mouse IgG | Cytiva | Cat# NA931 |
| AlexaFluor488-conjugated polyclonal donkey anti-mouse IgG | Thermo Fisher Scientific | Cat# A21202 |
| AlexaFluor647-conjugated polyclonal donkey anti-sheep IgG | Thermo Fisher Scientific | Cat# A21448 |
| AlexaFluor568-conjugated OROV NpE3 nanobody | This study | |
| AlexaFluor568-conjugated OROV SpE8 nanobody | This study | |
| **Oligonucleotides and other sequence-based reagents** | | |
| **Chemicals, enzymes and other reagents** | | |
| Freestyle MAX transfection reagent | Gibco | Cat# 16447100 |
| Freestyle 293 Expression Medium | Gibco | Cat# 12338018 |
| Geneticin | Gibco | Cat# 10131035 |
| Dulbecco's Modified Eagle Medium, high glucose, pyruvate powder | Gibco | Cat# 1280017 |
| Penicillin-Streptomycin Solution | Corning | Cat# 30-002-CL |
| Antibiotic-Antimycotic (100X) Penicillin, Streptomycin and Amphotericin B | Gibco | Cat# 15240062 |

| Reagent/resource | Reference or source | Identifier or catalogue number |
|---|---|---|
| Carboxymethylcellulose | Sigma-Aldrich | Cat# C9481 |
| Isopropyl β-D-thiogalactopyranoside | Melford | Cat# 2946-I56000 |
| 25 kDa branched polyethylenimine | Sigma-Aldrich | Cat# 408727 |
| Bovine DNase I | Merck | Cat# D4527 |
| EDTA-free protease inhibitor cocktail | Merck | Cat# P8849 |
| Benzonase nuclease | Merck | Cat# E1014 |
| Ni-NTA agarose | Qiagen | Cat# 30230 |
| HiTrap Protein G affinity column | Cytiva | Cat# 29048581 |
| HiTrap Mab select Prism A affinity column | Cytiva | Cat# 17549851 |
| HiLoad 16/600 Superdex 75 size-exclusion chromatography column | Cytiva | Cat# 28989333 |
| Superdex 200 10/300 GL size-exclusion chromatography column | Cytiva | Cat# 28906561 |
| Glutathione Sepharose 4B | Cytiva | Cat# 17075605 |
| Streptavidin | Leinco Technologies | Cat# S203 |
| AlexaFluor568-Maleimide | Thermo Fisher Scientific | Cat# A20341 |
| Zeba dye and biotin removal spin columns | Thermo Fisher Scientific | Cat# A44296 |
| EZ-Link plus activated peroxidase kit | Thermo Fisher Scientific | Cat# 31489 |
| PD minitrap G-10 columns | Cytiva | Cat# 28918010 |
| 96-well clear flat-bottom polystyrene high bind microplates | Corning | Cat# 9018 |
| 3,3′,5,5′-tetramethylbenzidine | Merck | Cat# 860336 |
| HRP-conjugated streptavidin | Thermo Fisher Scientific | Cat# N100 |
| Neutravidin | Thermo Fisher Scientific | Cat# A2666 |
| Streptavidin (SA) biosensors | Satorius | Cat# 18-5019 |
| Fluoromount-G mounting medium | Invitrogen | Cat# 00-4958-02 |
| Bio-Rad Protein Assay Dye Reagent Concentrate | Bio-Rad | Cat# 5000006 |
| 0.45 μm Nitrocellulose Membrane | Bio-Rad | Cat# 1620115 |
| **Software** | | |
| Prism v7.04 | GraphPad Software | |
| Octet Data Acquisition Software V6.4.1.150 | FortéBio | |
| Octet Data Acquisition Software V6.4.0.24 | FortéBio | |
| ImageLab Software v6.0.1 | Bio-Rad | RRID:SCR_014210 |
| **Other** | | |
| TS series cell disruptor | Constant Systems | |

| Reagent/resource | Reference or source | Identifier or catalogue number |
|---|---|---|
| SpectraMax i3 multimode microplate reader | Molecular Devices | |
| Varioskan LUX multimode microplate reader | Thermo Fisher Scientific | |
| Octet RED | FortéBio | |
| LSM 780 confocal microscope | Zeiss | |
| Chemidoc MP Imaging System | Bio-Rad | Cat# 731BR03509 |

## Plasmids

The N protein (GenBank AJE24680.1) and Gc spike region (residues 482–894) of the M polyprotein (GenBank AJE24679.1) from OROV strain BeAn19991 (Acrani et al, 2015), and the Gc spike region (residues 478–906) of the M polyprotein from Cristoli virus (CRIV; GenBank MN488997.1) (Rodriguez et al, 2020), were ordered as codon-optimised synthetic genes (GeneArt). OROV N was cloned into pOPTH, derived from pOPT (Teo et al, 2004) and encoding an N-terminal $His_6$ tag, or into pMWCAVI, encoding an N-terminal $His_6$ tag, human rhinovirus 3 C protease site and biotin acceptor peptide (AviTag). The OROV and CRIV Gc spikes with N-terminal secretion signals and C-terminal $His_6$ tags were cloned into the vector PB-TSW, which is derived from PB-T-PAF (Li et al, 2013) after the insertion of a woodchuck hepatitis virus (WHV) posttranscriptional regulatory element (WPRE) at the 3′ end. The pF5A_PBase plasmid encoding the piggyBac transposase (PBase) was generated by subcloning the PBase sequence from pPBase (Li et al, 2013) into the vector pF5A (Promega). The PB-RN plasmid carrying the reverse tetracycline transactivator inducer and neomycin resistance gene was previously described (Li et al, 2013). A second OROV Gc spike construct was generated with a C-terminal AviTag plus $His_6$ tag. Nanobodies with an N-terminal PelB periplasmic targeting sequence and C-terminal $His_8$ tag were generated by inverse PCR and blunt-end ligation of the original clones, generated in pADL-23c. For in vitro biotinylation of nanobodies, an AviTag sequence was added ahead of a $His_9$ tag using Gibson assembly. For AlexaFluor568 conjugation, a cysteine residue was added before the $His_8$ tag by QuikChange mutagenesis. For expression of nanobody-Fc fusions, VHH sequences were cloned into pHLSec (Aricescu et al, 2006) encoding the heavy chain of human IgG1, with the VHH domains cloned before the IgG hinge region. For co-translational biotinylation, an AviTag was added to the C terminus of the IgG1 Fc3 domain using inverse PCR and a secreted BirA-FLAG plasmid (Bushell et al, 2008) was used. Plasmids encoding the proteins described in this study, and selected cell lines plus purified proteins, will be made available from Addgene and/or the Centre for Infectious Disease Reagents (https://nibsc.org/science_and_research/idd/cfar.aspx), UK Medicines and Healthcare products Regulatory Agency.

## Nanobody generation

Nanobodies were generated as described previously (Eyssen et al, 2024). Briefly, N and Gc spike (200 μg each) were mixed with the

adjuvant Gerbu LQ#3000 for each of three intramuscular immunisations of a llama on days 0, 28 and 56. Blood (150 mL) was collected on day 66. Immunisations and handling of the llama were performed under the authority of the project license PA1FB163A. VHHs were amplified by two rounds of PCR from cDNA prepared from peripheral blood monocytes and cloned into the SfiI sites of the phagemid vector pADL-23c (Antibody Design Laboratories, San Diego, USA). Electro-competent *Escherichia coli* TG1 cells (Agilent) were transformed with the recombinant pADL-23c vectors, and the resulting TG1 library stock was infected with M13K07 helper phage to obtain a library of VHH-presenting phage. Phage displaying VHHs specific for N and Gc spike were enriched after two rounds of bio-panning on 50 nM and 5 nM of biotinylated protein, respectively, through capturing with Dyna-beads M-280 (Thermo Fisher Scientific). After the second round of panning, 93 individual phagemid clones were picked, VHH-displaying phage were recovered by infection with M13K07 helper phage and tested for antigen binding by ELISA with biotin-tagged N or Gc spike immobilised on neutravidin-coated plates. Positive phage binders were sequenced and grouped according to CDR3 sequence identity using the IMGT/V-QUEST server (Giudicelli et al, 2011).

## Mammalian cell culture

Mycoplasma-free Freestyle 293F suspension cells (Thermo Fisher Scientific) were grown in Freestyle 293F medium (Gibco) on a shaking platform (125 rpm) in a humidified 8% $CO_2$ atmosphere at 37 °C. Cells stably expressing OROV Gc Spike and CRIV Gc Spike under control of the tetracycline response element promoter were generated using the PiggyBac transposase system as previously described (Li et al, 2013). Briefly, A 30 mL suspension culture of Freestyle 293F cells at $1 \times 10^6$ cells/mL was transfected with a 5:1:1 mass ratio of PB-TSW(Gc spike): PB-RN: PBase (35 μg total DNA) using Freestyle MAX transfection reagent (Gibco) as per the manufacturer's instructions. After 2 days, the cells were transferred to fresh media supplemented with 500 μg/mL geneticin (Gibco), and the drug selection was continued for 2 weeks with media replenishment every 3 days.

Mycoplasma-free HeLa and Vero cells were obtained from ATCC (American Type Culture Collection) and cultured in Dulbecco's Modified Eagle Medium (DMEM; Gibco) supplemented with 10% (w/v) foetal bovine serum (FBS), 100 U/mL penicillin and 100 μg/mL streptomycin at 37 °C in a humidified 5% $CO_2$ atmosphere. Cell lines have not been independently authenticated by short tandem repeat profiling.

## Protein expression

OROV N was expressed in *E. coli* T7 Express *lysY/I^q* cells (New England Biolabs) in 2×TY medium at 37 °C to an $OD_{600}$ of 0.6. The temperature was dropped to 22 °C and protein expression by the addition of 0.4 mM isopropyl β-D-thiogalactopyranoside (IPTG). The following morning, cells were harvested by centrifugation and stored at −80 °C until required.

For OROV and CRIV Gc Spike, stably transfected cells at $0.8–1.0 \times 10^6$ cells/mL were induced by addition of 0.5–2 μg/mL doxycycline and cultured for 48–72 h before harvesting of cell pellets. Supernatant was 0.2-μm filtered and stored at 4 °C until required.

Nanobodies were expressed in *E. coli* WK6 cells (ATCC) in Terrific Broth supplemented with 2% (v/v) glucose at 37 °C to an $OD_{600}$ of 0.8–1.2. Protein expression was induced by adding 1 mM ITPG, and the temperature was lowered to 28 °C. The following morning, cells were harvested by centrifugation and used immediately for protein purification.

For nanobody-Fc fusions, Freestyle 293F cells at $1 \times 10^6$ cells/mL were transiently transfected by the addition of DNA that had been pre-incubated in PBS with 25 kDa branched polyethylenimine (PEI), adding 1 μg DNA and 1.5 μg PEI per mL of cultured cells. For nanobody-Fc fusions with AviTags, the plasmid encoding secreted BirA-FLAG was co-transfected with the plasmid encoding the nanobody-Fc fusion at 1:9 DNA ratio, and the culture was supplemented with 100 μM D-biotin. Cells were cultured for a further 48 h before being harvested. Supernatant was 0.2-μm filtered and stored at 4 °C until required.

## Protein purification

For OROV N, pellets were resuspended at 4 °C in Ni-NTA lysis buffer (25 mM Tris pH 7.5, 500 mM NaCl, 20 mM imidazole, 0.5 mM $MgCl_2$, 1.4 mM β-mercaptoethanol, 0.05% TWEEN-20) supplemented with 200 U bovine DNase I (Merck), 200 μL EDTA-free protease inhibitor cocktail (Merck) and 500 U benzonase (Merck) before lysis using a TS series cell disruptor (Constant Systems) at 24 kpsi. Lysates were cleared by centrifugation (40,000 × *g*, 30 min, 4 °C) and incubated with Ni-NTA agarose (Qiagen) for 1 h at 4 °C before extensive washing (≥20 column volumes [c.v.]) of Ni-NTA wash buffer (20 mM Tris pH 7.5, 500 mM NaCl, 20 mM imidazole) and elution using Ni-NTA elution buffer (20 mM Tris pH 7.5, 500 mM NaCl, 250 mM imidazole).

For OROV and CRIV Gc spike, the filtered supernatant was supplemented with 25 mM Tris pH 8.5, 300 mM NaCl and 0.5 mM $MgCl_2$ before being incubated with Ni-NTA agarose (Qiagen) for 1 h at 4 °C. The Ni-NTA agarose was washed with ≥10 c.v. of 20 mM Tris pH 8.0, 300 mM NaCl and proteins were eluted using PBS supplemented with 250 mM imidazole pH 7.5.

For nanobodies, the pellet from 1 L of bacterial culture was resuspended in 15 mL cold TES buffer (200 mM Tris pH 8.0, 500 mM sucrose, 0.5 mM EDTA) before being diluted threefold in cold TS buffer (50 mM Tris pH 8.0, 125 mM sucrose) supplemented with 200 U bovine DNase I (Merck) and 200 μL EDTA-free protease inhibitor cocktail (Merck) to lyse the periplasm. After 30 min incubation at 4 °C, cells were harvested by centrifugation at 30,000 × *g*. The supernatant was diluted fivefold with cold PBS supplemented with 0.5 mM $MgCl_2$ before being incubated with Ni-NTA agarose for 30–60 min at 4 °C. The Ni-NTA agarose was washed with ≥10 c.v. of PBS supplemented with 10 mM imidazole pH 7.5, and nanobodies were eluted using PBS supplemented with 250 mM imidazole pH 7.5.

For nanobody-Fc fusions, the supernatant was stabilised by addition of a 100 mM sodium phosphate pH 7.0 at a 6:1 supernatant:buffer ratio. The supernatant was then injected onto HiTrap Protein G or Mab select Prism A columns (Cytiva) that were washed with 20 mM sodium phosphate pH 7.0 before the protein was eluted using 0.1 M glycine pH 2.7. The pH of fractions was

immediately neutralised via elution into collection vials containing 1 M Tris pH 8.5.

All proteins were subsequently subjected to size-exclusion chromatography using Superdex 75 16/600 (bacterial expression) or 10/300 (mammalian expression) columns (Cytiva) equilibrated with phosphate-buffered saline (PBS).

## Protein labelling

With the exception of nanobody-Fc fusions, proteins with an AviTag were biotinylated in vitro using GST-tagged *E. coli* BirA as described in (Fairhead and Howarth, 2015). Briefly, 100 μM Avi-Tagged nanobody was incubated with 5 mM MgCl$_2$, 2 mM ATP, 150 μM D-biotin and 1 μM GST-BirA at 30 °C for 1 h. The reaction was supplemented with a further 150 μM D-biotin and 1 μM GST-BirA before incubation at 30 °C for a further 1 h. The GST-BirA was removed by depletion using glutathione Sepharose 4B (Cytiva), and excess biotin was removed via dialysis. The extent of biotinylation was assayed via a streptavidin shift assay, where 0.5-, one- and twofold molar excess of purified streptavidin (Leinco Technologies) was added to samples of biotinylated protein that had been boiled in SDS-PAGE loading buffer. The electrophoretic mobility of streptavidin-bound biotinylated nanobody is dramatically altered in SDS-PAGE. This allowed determination of percentage biotinylation, via calculating the ratio of unshifted protein in samples lacking streptavidin versus samples with twofold molar excess of streptavidin.

For AlexaFluor568 labelling, 500 μg of Cys-His$_8$ tagged nanobodies in 100 μL PBS were supplemented with 2.5 mM TCEP and incubated at 4 °C for 10 min before addition of 8.2 mM AlexaFluor568-Maleimide (Thermo Fisher Scientific). The reaction was incubated in the dark at 4 °C for 1 h before unbound dye was removed using Zeba dye and biotin removal spin columns (Thermo Fisher Scientific) according to the manufacturer's instructions. AlexaFluor-labelled nanobodies were diluted to 1 mg/mL with PBS supplemented with 0.02% sodium azide and stored at 4 °C.

Horseradish peroxidase (HRP) conjugation was performed using the EZ-Link plus activated peroxidase kit (Thermo Fisher Scientific) according to the manufacturer's instructions. FcSpB6 and FcSpC7 were conjugated in PBS, pH 7.4 using a 2:1 molar ratio of activated HRP to nanobody-Fc fusion. FcNpD2 was conjugated in 0.2 M carbonate-bicarbonate buffer, pH 9.4, using a 4:1 molar ratio of HRP to nanobody-Fc fusion. Following HRP conjugation, nanobody-Fc fusions were exchanged into PBS using a PD minitrap G-10 column (Cytiva), mixed 1:1 with 100% (v/v) glycerol and stored at −20 °C.

## Virus preparation

Unless otherwise indicated, the OROV strain used in this study was BeAn19991, kindly provided by Prof. Luiz Tadeu Moraes Figueiredo (University of São Paulo). For animal infection, ELISA and in vitro neutralisation assays, OROV BeAn19991 (Acrani et al, 2015; Aquino et al, 2003), TRVL (Anderson et al, 1961) and AM0088 (Scachetti et al, 2025) stocks were prepared in Vero cells, which were infected when they reached 90–95% confluence. The cultures were maintained until approximately 80% of the cells exhibited a cytopathic effect. The supernatant was then collected and clarified by centrifugation at 5000 × *g* for 10 min at 4 °C. Titration was performed in Vero cells infected with serial tenfold

dilutions of the stocks, which were maintained in semi-solid medium containing 0.75% carboxymethylcellulose (CMC). After 72 h the cells were fixed with 4% (v/v) formaldehyde and stained with 1% (w/v) crystal violet for plaque counting. The titres obtained were $9.0 \times 10^6$ PFU/mL for the BeAn19991 strain, $1.4 \times 10^5$ pfu/mL for TRVL, and $1.0 \times 10^6$ PFU/mL for the AM0088 reassortant.

For immunocytochemistry and immunoblotting, virus stocks were produced via intercranial injection of neonatal C57BL/6 mice. The brain was homogenised when neurological sequelae became apparent, 2–3 days post-infection. For OROV BeAn19991, the homogenate was used to inoculate Vero cells. At 36 h post-infection (hpi), Vero cell supernatants were clarified by centrifugation (3000 × *g* for 10 min), frozen and stored at −80 °C. For OROV AM0088, the homogenate was diluted in DMEM and used directly for infections.

## Animals and infections

Animal experiments were conducted according to the ethical standards of the local ethics committee (CETEA) of the Ribeirão Preto School of Medicine, University of São Paulo, under protocol number 1366/2024R1. Food and water were provided ad libitum, and four animals were housed per cage, each cage containing wood shavings and other environmental enrichment materials. For monitoring immune response, six-week-old male C57BL/6 mice were infected subcutaneously with $10^6$ PFU of OROV, and blood was collected by facial vein puncture on days 7, 14 and 21 post-infection. For immunisations, 6-week-old male C57BL/6 mice were immunised with intramuscular injections containing 35 μg of OROV Gc spike or a combination of 35 μg OROV Gc spike and 30 μg OROV N, both with incomplete Freund's adjuvant. Three immunisations were administered, with two-week intervals between them. Prior to the booster immunisation, and 6 weeks after the final boost, a total blood sample was collected by puncture of the facial vein. The serum was separated by centrifugation at 3000 rpm for 10 min at room temperature before being used for neutralisation and ELISA assays. Mice were randomly assigned to experimental versus control groups, and sample sizes were selected with reference to previous studies (Stubbs et al, 2021).

## Enzyme-linked immunosorbent assays (ELISAs)

All ELISA assays were performed using 96-well clear flat-bottom polystyrene high bind microplates (Corning 9018) using PBS plus 0.05% TWEEN-20 (PBS-T) as wash buffer and PBS-T supplemented with 2% (w/v) bovine serum albumin (BSA) as blocking and antibody/antigen dilution buffer. Plates were washed at least four times with PBS-T between incubation or detection steps. Assays were developed using 150 μg/mL 3,3′,5,5′-tetramethylbenzidine (TMB), 0.005% H$_2$O$_2$ (approx. 1.6 mM) in 0.1 M sodium acetate pH 5.0 for 30 min before mixing with an equal volume of 0.16 M sulfuric acid and measuring absorbance at 450 nm. All incubations were performed at room temperature unless otherwise stated.

For indirect ELISAs to monitor for antibodies against OROV mice, microplates were coated with purified antigen (2 μg/mL in PBS) overnight at 4 °C. A standard curve was generated by coating the plates with increasing concentrations of purified mouse IgG (Merck I5381) or IgM (Thermo Fisher Scientific MA1-10438). Plates were blocked for 2 h before being incubated with serum diluted in blocking buffer (1 in 250,000 for

IgG and 1 in 200,000 for IgM [infected animals], or 1 in 800,000 for IgG [immunised animals]). Plates were subsequently incubated with isotype-specific HRP-conjugated secondary antibodies (goat anti-mouse IgG, 1:10,000 dilution, Thermo Fisher Scientific 31430, and goat anti-mouse IgM, 1:2000 dilution, Merck AP128P) for 30–60 min before developing.

For indirect ELISA to assess binding of biotinylated nanobodies to purified antigens, microplates were coated with purified antigen (OROV N, OROV Gc spike or CRIV Gc spike; 10 μg/mL in PBS) for 2 h and blocked overnight. Plates were incubated with biotinylated nanobodies (2 μg/mL in blocking buffer) for 1 h, HRP-conjugated streptavidin (1:10,000 dilution, Thermo Fisher Scientific N100) for 15 min, and then developed.

For sandwich ELISAs using nanobodies, microplates were coated with His$_8$-tagged nanobodies (2 μg/mL in PBS) for 2 h and blocked overnight at room temperature or 4 °C. Plates were then incubated with purified antigens in blocking buffer (OROV N or OROV Gc spike, 10 μg/mL or diluted as shown for standard curves, or virus stocks diluted as shown) for 1 h, biotinylated nanobody (2 μg/mL in blocking buffer) for 30–60 min, HRP-conjugated streptavidin (1:10,000 dilution) for 15 min, and then developed.

For sandwich ELISAs using nanobody-Fc fusions, microplates were coated with neutravidin (10 μg/mL in PBS, Fisher Scientific A2666) for at least overnight at 4 °C. Plates were incubated with capture reagent (biotinylated nanobody-Fc fusion, 2 μg/mL in blocking buffer) for 1 h, blocked for 1 h, incubated with antigen in blocking buffer (serum diluted 1 in 4, virus stocks diluted as shown, or purified antigens diluted as shown for standard curves) overnight at 4 °C, incubated with detection reagent (HRP-conjugated nanobody-Fc fusion, 0.4 μg/mL in blocking buffer), then developed.

For calculating limits of detection for sandwich ELISAs, standard curves of antigen were generated using purified antigen and the standard deviation of the background was calculated from at least four wells to which no antigen was added. The limit of detection is calculated as 3.3 * standard deviation of background / slope of standard curve (Girt et al, 2021).

RT-qPCR analysis of clinical samples was performed as described previously (Naveca et al, 2017). Least squares non-linear regression of the $log_{10}$-transformed Z scores and Ct values was performed using Prism 7 (GraphPad Software).

## Biolayer interferometry (BLI)

BLI experiments were performed at 30 °C in PBS-T with plate shaking (1000 rpm) using an Octet Red instrument (FortéBio). Streptavidin biosensors (Satorius) were hydrated in PBS-T for at least 10 min before use. Sensors were loaded with biotinylated nanobodies at 15 μg/mL (bNpG1) or 25 μg/mL (all others) for 210 s. Reference subtraction was performed using a sensor that had not been loaded with biotinylated nanobody. Sensors were regenerated using two cycles of 10 s immersion in 10 mM glycine pH 2.1 followed by 20 s neutralisation in PBS-T before use and between association phases. Sensors were incubated with PBS-T for 120 s (baseline) before association via incubation with 1 μM OROV N in the presence or absence of 25 μM competing nanobody for 300 s. For each biosensor, the baseline-corrected reference-subtracted signal at 180 s in the presence of the competing nanobody was divided by the signal for the same biosensor at 180 s in the absence of the competitor to determine fold change in association. Reference and baseline subtraction, Savitzky-Golay Filtering and report point generation were performed using Octet Data Analysis software (FortéBio).

## Neutralisation assays

For neutralisation using nanobody-Fc fusions or animal serum, the samples were twofold serially diluted with PBS (from 2 μM for nanobody-Fc fusions and from 1:25 for serum) before being incubated with an equal volume of virus particles for 1 h at 37 °C. The inoculum was added to Vero cells at 95% confluence, and the cultures were incubated at 37 °C with gentle manual agitation every 15 min. After 1 h, the inoculum was removed and semi-solid medium (DMEM with 2% (w/v) FBS, 100 U/mL penicillin, 100 μg/mL streptomycin, 0.25 μg/mL Amphotericin B and 0.75% CMC) was added. Staining was performed 72 hpi using 1% (w/v) crystal violet, following fixation with 4% (v/v) formaldehyde for 30 min. Plaques were counted, and neutralisation was assessed by evaluating the infection of OROV at each nanobody concentration relative to a PBS control. The neutralising dose ($ND_{50}$) was calculated by non-linear fitting to a three-parameter inhibitor-response curve where the top and bottom values were constrained to 100 and 0, respectively, using Prism 7 (GraphPad Software).

## Immunocytochemistry

Confluent monolayers of HeLa were infected by the addition of OROV diluted in serum-free media (MOI 0.5) and rocking the cells on ice for 1 h. Cells were moved to 37 °C for 15 min, washed with PBS, and then overlain with DMEM with 2% (w/v) FBS. At 20 hpi, cells were washed with PBS and fixed for 10 min with 4% (v/v) paraformaldehyde in PBS. Cells were permeabilised for 30 min using 0.1% saponin in blocking buffer (0.2% porcine gelatine in PBS). Permeabilised cells were incubated with primary antibodies and anti-OROV nanobodies at 37 °C for 30 min, diluted as follows: mouse anti-OROV antiserum, 1:500 (Concha et al, 2024); AlexaFluor568 tagged anti-OROV nanobodies, 1:4000; anti-TGN46 (Bio-Rad AHP500), 1:500. After washing, secondary antibodies were applied at 37 °C for 30 min, diluted as follows: AlexaFluor488 anti-mouse and AlexaFluor647 anti-sheep (Thermo Fisher Scientific), 1:1000. Cells were mounted on glass slides using Fluoromount-G and imaged at ×40 magnification using a Zeiss LSM 780 confocal microscope at the Ribeirão Preto Medical School Multiuser Laboratory of Multiphoton Microscopy.

## Immunoblotting

Confluent monolayers of HeLa cells in six-well plates were washed with PBS and then infected by the addition of OROV (MOI 5) with gentle rocking on ice for 90 min before transfer to 37 °C for 15 min to facilitate virus entry. Cells were washed and incubated in DMEM with 2% (w/v) FBS at 37 °C. At 24 hpi, cells were washed, detached by incubation with 2 mM EDTA in PBS for 5 min, harvested by centrifugation at 2000 × $g$ for 10 min at 4 °C, and the cell pellets were stored at −80 °C. Cells were resuspended and incubated on ice for 20 min in lysis buffer comprising 50 mM Tris pH 7.5, 150 mM NaCl, 10% (v/v) glycerol, 5 mM EDTA, 1% (v/v) Triton X-100, plus protease inhibitors (P8340; Sigma-Aldrich). Lysates were clarified (16,000 × $g$, 10 min, 4 °C), protein concentration was equalised following analysis using the Bradford assay (Bio-Rad), and samples

**The paper explained**

**Problem**

Oropouche virus (OROV) causes Oropouche Fever, a debilitating febrile illness endemic to the Amazon basin. Since late 2023, there has been an epidemic of OROV infection caused by a new strain of the virus, with the first recorded deaths in healthy adults and infection of pregnant women leading to foetal loss. Historically, OROV has been understudied and, as a result, we have only limited molecular tools for detecting and studying OROV infection.

**Results**

Here, we report a protein-based molecular toolkit for OROV. Purified synthetic OROV components (antigens) enable detection of prior OROV infection in mice, and immunisation of mice with these antigens elicits an antibody response that can block OROV infection in cultured cells. We generated antibody-like detection reagents (nanobodies) that bind OROV antigens and developed a diagnostic assay using these nanobodies that detects acute human OROV infection. Nanobodies that bind a protein on the surface of OROV potently block OROV infection of cultured cells, and these nanobodies can also be used for laboratory research into the molecular characteristics of OROV infection.

**Impact**

The tools presented here will accelerate OROV research and diagnosis. The ability to produce high-quality OROV antigens at scale will enable serological surveys to assess the prevalence of prior OROV infection in the community. Purified OROV antigens also show promise as candidate vaccine molecules. The antigen detection assay presented here establishes proof-of-concept that OROV proteins can be detected in the blood of individuals suffering from acute OROV infection. With suitable optimisation, it may be possible to develop a point-of-care rapid antigen detection (lateral flow) test using these nanobodies to accelerate OROV diagnosis in low-resource settings.

were diluted in Laemmli buffer supplemented with β-mercaptoethanol. Samples were heated to 95 °C for 5 min, separated by SDS-PAGE and transferred to 0.45 μm nitrocellulose membranes (Bio-Rad). Membranes were blocked using 5% (w/v) skim milk powder in PBS with 0.1% TWEEN-20 for 1 h before being incubated with either mouse anti-OROV antiserum (1:500), then HRP-conjugated sheep anti-mouse (1:1000, Cytiva), or with HRP-conjugated nanobody-Fc fusions FcNpD2, FcSpB6 or FcSpC7 (1:1000). Protein bands were visualised using enhanced chemiluminescence reagents and the ChemiDoc Imaging System with ImageLab software (Bio-Rad).

## Ethics statement

Immunisations and handling of llama were performed under the authority of UK Home Office project license PA1FB163A. All mouse experiments were approved by the local ethics committee at the Ribeirão Preto School of Medicine (CETEA), under protocol number 1366/2024R1. Informed consent was obtained from all human subjects and confirm that the experiments conformed to the principles set out in the WMA Declaration of Helsinki and the Department of Health and Human Services Belmont Report. Use of human clinical samples was approved by the Fundação de Medicina Tropical Dr. Heitor Vieira Dourado ethics committee under registration number 700.915 and by the ethics committee of the Institutio Esperança de Ensino Superior under registration number 3.951.431.

## Data availability

This study includes no data deposited in external repositories.

The source data of this paper are collected in the following database record: biostudies:S-SCDT-10_1038-S44321-025-00291-7.

## Peer review information

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

## Acknowledgements

We thank Bernard Kelly (University of Cambridge) for the pMWCAVI plasmid and DNA encoding human IgG1, and Prof. Luis Tadeu Moraes Figueiredo (University of São Paulo) for the kind gifts of OROV BeAn19991, TRVL and anti-OROV mouse antiserum. We acknowledge the Ribeirão Preto Medical School Multiuser Laboratory of Multiphoton Microscopy for access to the Zeiss LSM 780 confocal microscope. This work was supported by a Biotechnology and Biological Sciences Research Council (BBSRC) grant to RJO (BB_V018523), an International Collaboration Award from the Royal Society to EA and SCG (ICA\R1\201019), a São Paulo Research Foundation (FAPESP) grant to EA (19/26119-0), and International Science Partnerships Fund (ISPF) Institutional Support Grant (ODA) from Research England to SCG. The funders had no role in study design, data collection and analysis, decision to publish, or preparation of the manuscript. For the purpose of open access, the author has applied a Creative Commons Attribution (CC BY) licence to any Author Accepted Manuscript version arising.

## Author contributions

**Monique K Merchant**: Investigation; Visualisation; Writing—review and editing. **Juliano de Paula Souza**: Investigation; Visualisation; Writing—review and editing. **Sahar Abdelkarim**: Investigation. **Shrestha Chakraborty**: Investigation. **Tufi A Nasser Neto**: Investigation. **Kristel I Gutierrez Manchay**: Investigation; Visualisation; Writing—review and editing. **Daniel M de Melo Jorge**: Investigation. **Valdinete Alves do Nascimento**: Investigation. **Yuxian Sun**: Investigation. **Eve R Caroe**: Investigation. **Lauren E A Eyssen**: Investigation. **Jared S Rudd**: Investigation; Writing—review and editing. **Isa C Ribeiro Piaulino**: Investigation. **Sérgio Damasceno Pinto**: Investigation. **Matheus H Pereira da Silva**: Investigation. **Felipe Rocha do Nascimento**: Investigation. **Felipe Gomes Naveca**: Resources. **José L Proenca-Modena**: Resources; Writing—review and editing. **Luis L P da Silva**: Supervision; Writing—review and editing. **Regina M Pinto de Figueiredo**: Resources; Writing—review and editing. **Raymond J Owens**: Supervision; Funding acquisition; Project administration; Writing—review and editing. **Eurico Arruda**: Conceptualisation; Supervision; Funding acquisition; Project administration; Writing—review and editing. **Stephen C Graham**: Conceptualisation; Data curation; Supervision; Funding acquisition; Investigation; Visualisation; Writing—original draft; Project administration; Writing—review and editing.

Source data underlying figure panels in this paper may have individual authorship assigned. Where available, figure panel/source data authorship is listed in the following database record: biostudies:S-SCDT-10_1038-S44321-025-00291-7.

## Disclosure and competing interests statement

The authors declare no competing interests.

# Expanded View Figures

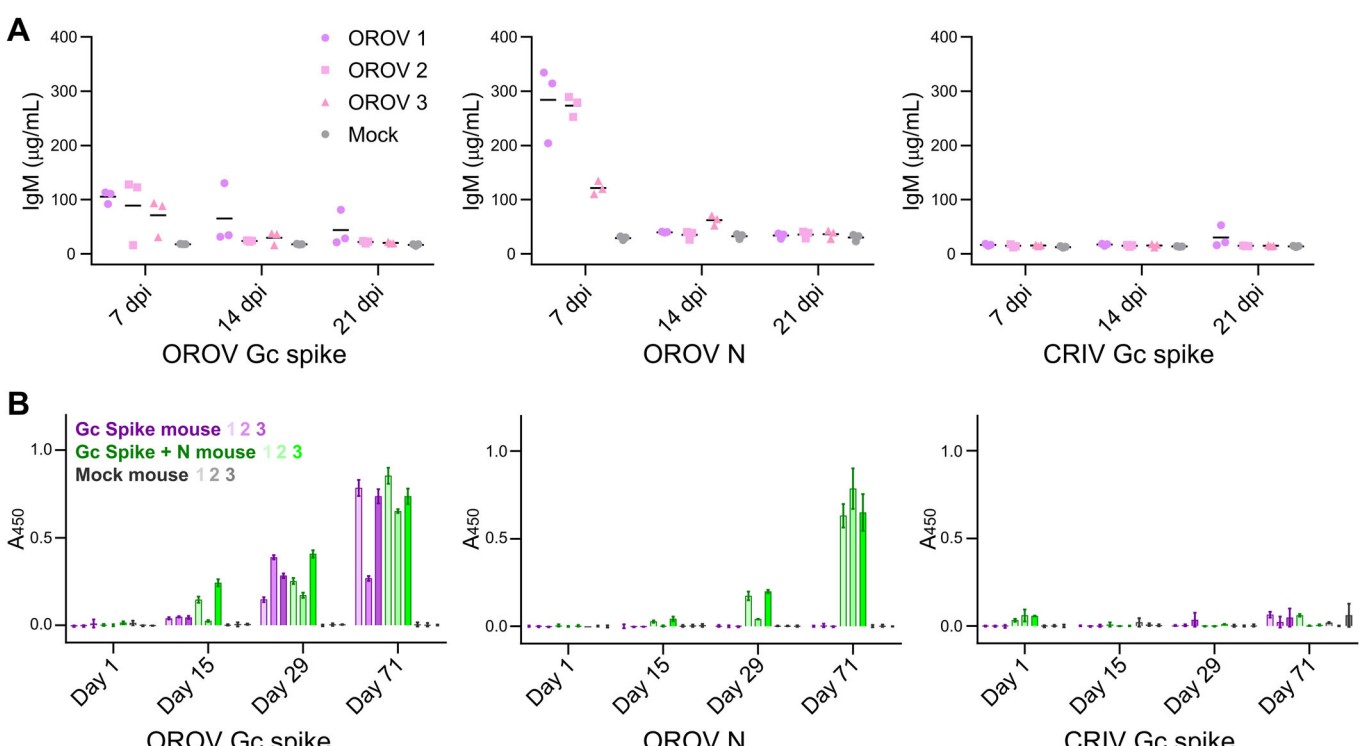

**Figure EV1.  Purified OROV antigens are recognised by antibodies raised in response to OROV infection, and purified antigens raise an immune response in mice.**

(A) Blood was harvested at indicated days post-infection (dpi) and an indirect ELISA to detect IgM was performed using indicated antigens. Values are show for three independent ELISA experiments using serum from three infected mice or one mock-infected control mouse. (B) Pre-boost serum was collected from mice immunised at day 1 and the boosted at days 15 and 29 with purified OROV antigens Gc spike alone (purple), OROV Gc spike plus N (green), or mock-immunised (3 mice per treatment). Serum was also collected 6 weeks after the final boost. The presence of antibodies that recognise OROV Gc spike, OROV N, or CRIV Gc spike (negative control) was tested by indirect ELISA. Mean ± SD of three independent measurements for each mouse is shown.

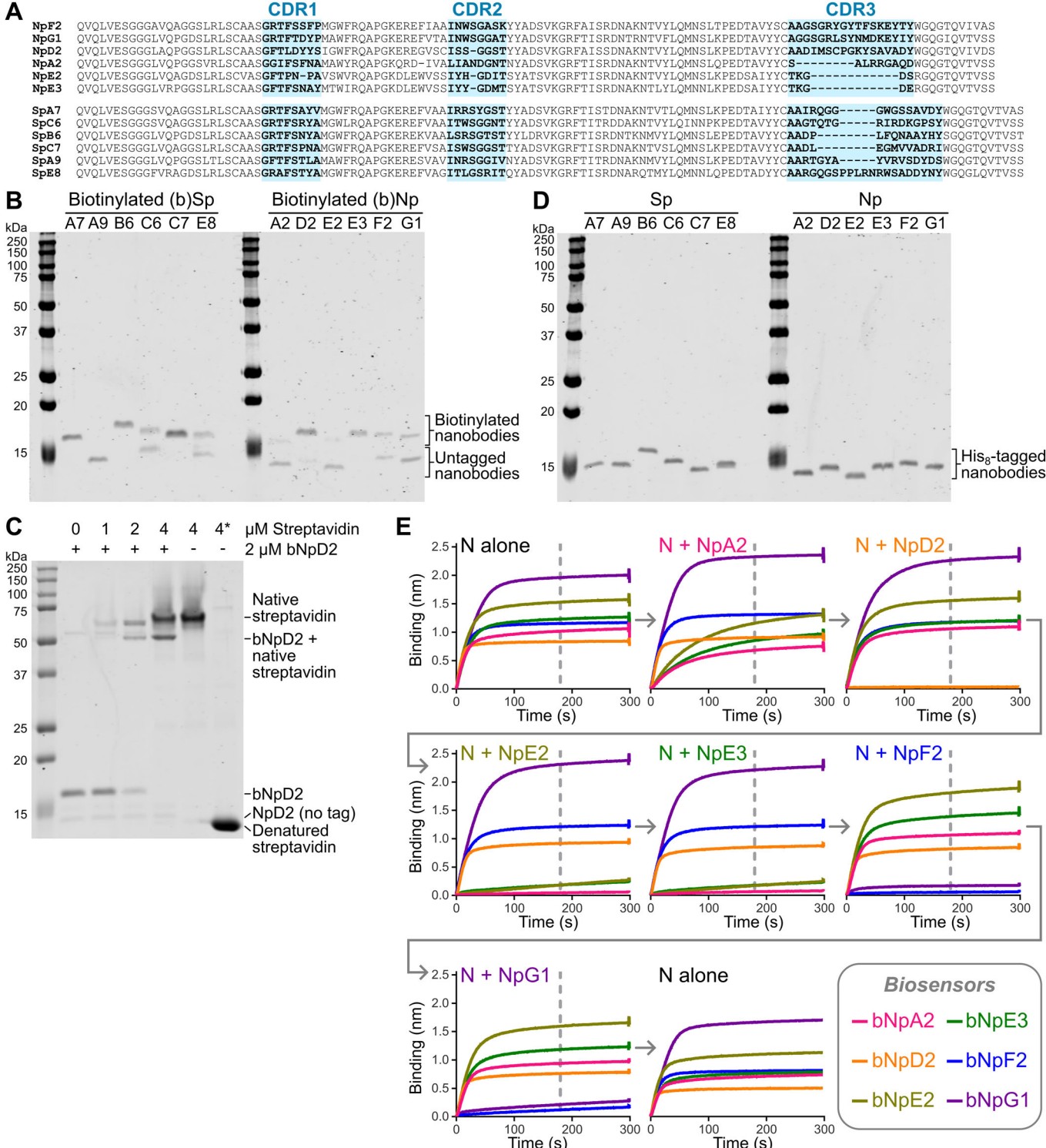

◀

**Figure EV2. Generation and purification of nanobodies to detect OROV Gc spike and N.**

(A) Amino acid sequence alignment of selected nanobodies that recognise OROV N (NpXX) or Gc spike (SpXX). Complementarity Determining Regions (CDRs) as defined by IGMT (Giudicelli et al, 2011) are highlighted. (B) Coomassie-stained SDS-PAGE of nanobodies that had been biotinylated in vitro. Upper bands represent the Avi-tagged, biotinylated nanobodies while lower bands represent nanobodies where the tag had been lost, presumably by proteolysis during the purification procedure. (C) Electrophoretic mobility shift assay to confirm biotinylation. After boiling of the nanobody in SDS-PAGE loading buffer, streptavidin was added at a 2:1, 1:1 or 1:2 molar ratio (nanobody:streptavidin) and samples were subjected to SDS-PAGE. Appearance of a high apparent molecular weight band, and disappearance of the low molecular weight band, confirms biotinylation of the nanobody. Asterisk (*) denotes streptavidin that was boiled before SDS-PAGE, rather than being added to the sample buffer after boiling. (D) Coomassie-stained SDS-PAGE of His$_8$-tagged nanobodies. (E) OROV N competition BLI sensorgrams. Streptavidin biosensors loaded with bNbs were sequentially incubated with 1 μM OROV N then with 1 μM OROV N plus 25 μM of each competitive nanobody, with biosensor regeneration between incubation (association) cycles. After the final competition step, the biosensor was incubated with 1 μM OROV N to confirm that the bNbs remained active. For each association in the presence of competitive nanobodies, the response for each sensor at 180 s (dotted grey line) was divided by the response of the same sensor in the presence of OROV N alone at 180 s to generate the heatmap shown in Fig. 2C. Source data are available online for this figure.

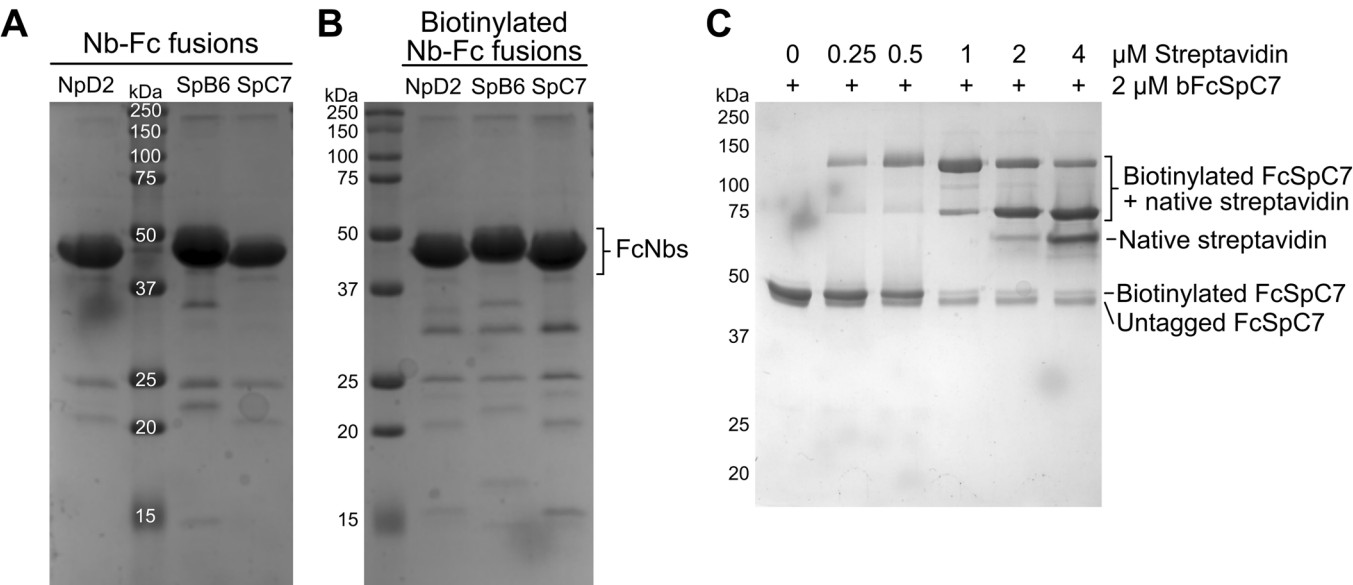

**Figure EV3.  Purification of nanobody-Fc fusions.**

(**A, B**) Coomassie-stained SDS-PAGE of (**A**) untagged nanobody-Fc fusions, and (**B**) biotinylated nanobody-Fc fusions. (**C**) Electrophoretic mobility shift assay to confirm biotinylation. After boiling of the nanobody-Fc fusion in SDS-PAGE loading buffer, streptavidin was added at a 8:1, 4:1, 2:1, 1:1 or 1:2 molar ratio (nanobody-Fc fusion:streptavidin) and samples were subjected to SDS-PAGE. Appearance of a high apparent molecular weight bands, and reduction of the lower molecular weight band, confirms biotinylation of the nanobody-Fc fusion.

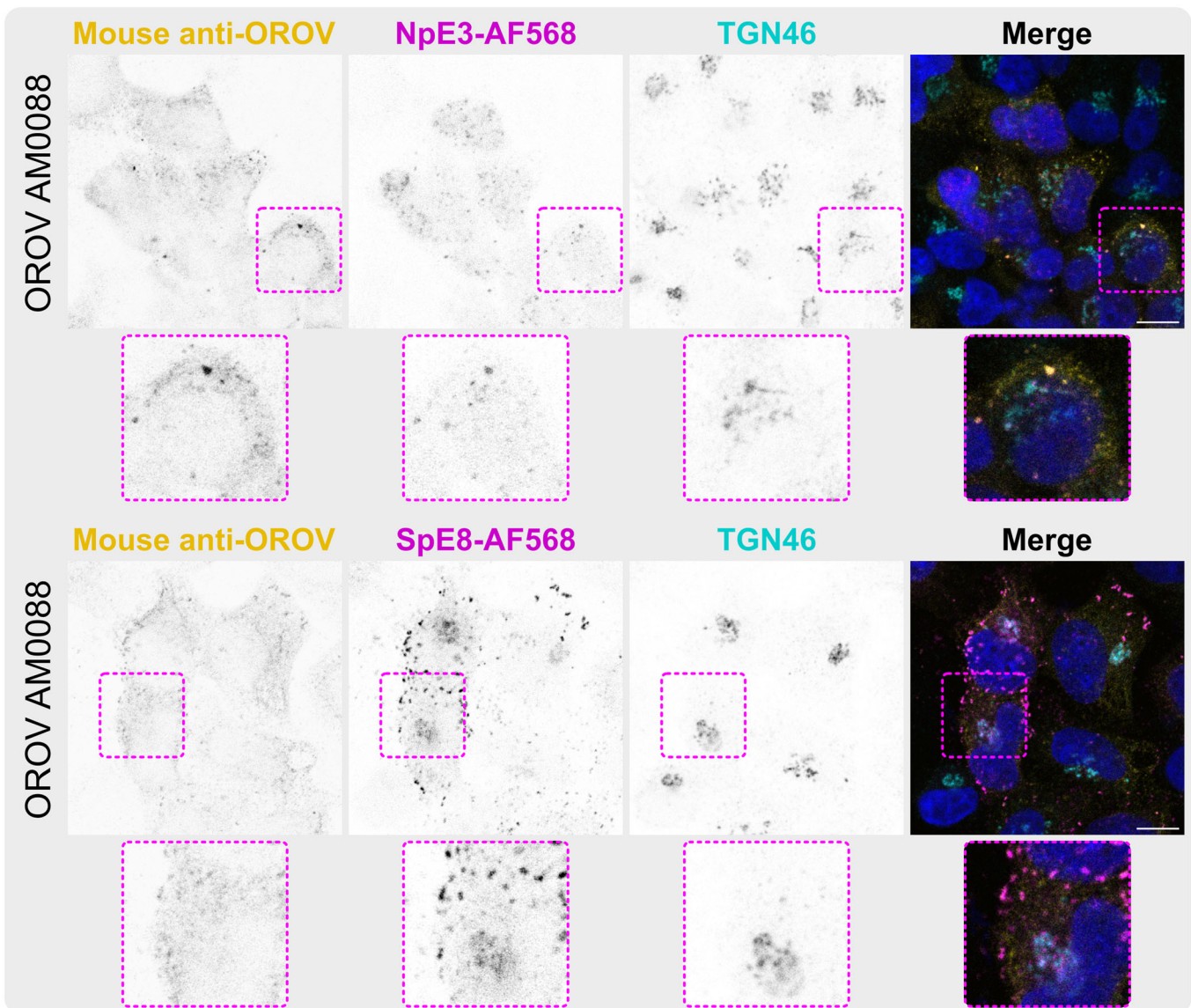

**Figure EV4.** **Nanobodies detect infection of HeLa cells with new OROV isolate AM0088.**

HeLa cells were infected with OROV AM0088 (MOI 0.5). Cells were probed with AlexaFluor (AF)568 conjugated OROV nanobodies against N (NpE3) or Gc Spike (SpE8), with a polyclonal antibody against OROV and with an antibody against TGN46, plus appropriate secondary antibodies. Nuclei are stained with DAPI (blue). Scale bar = 10 μm.

