## [Peer Review File · EMBO Molecular Medicine]

Protein-based tools for the detection and characterisation of Oropouche virus infection

Monique Merchant, Juliano de Paula Souza, Sahar Abdelkarim, Shrestha Chakraborty, Tufi Nasser, Kristel Gutierrez Manchay, Daniel de Melo Jorge, Valdinete Alves do Nascimento, Yuxian Sun, Eve Caroe, Lauren Eyssen, Jared Rudd, Isa Ribeiro Piauilino, Sérgio Damasceno Pinto, Matheus Pereira da Silva, Felipe Rocha do Nascimento, Felipe Gomes Naveca, José Luiz Proenca-Modena, Luis Silva, Regina Pinto de Figueiredo, Raymond Owens, Eurico Arruda, and Stephen Graham

Corresponding authors: Stephen Graham (scg34@cam.ac.uk) , Eurico Arruda (eaneto@fmrp.usp.br)

Review Timeline:

Submission Date:	9th Apr 25
Editorial Decision:	13th May 25
Revision Received:	30th Jun 25
Editorial Decision:	22nd Jul 25
Revision Received:	23rd Jul 25
Accepted:	25th Jul 25

Editor: Zeljko Durdevic

Transaction Report:

13th May 2025

Dear Dr. Graham,

Thank you for the submission of your manuscript to EMBO Molecular Medicine. We have now received feedback from the two reviewers who agreed to evaluate your manuscript. Both referees recognize interest of the study but also raise important concerns that should be addressed in a major revision. If you would like to discuss further the points raised by the referees, I am available to do so via email or video. Let me know if you are interested in this option.

We would welcome the submission of a revised version within three months for further consideration. Please let us know if you require longer to complete the revision.

I look forward to receiving your revised manuscript.

Yours sincerely,

Zeljko Durdevic

Zeljko Durdevic
Senior Editor
EMBO Molecular Medicine

We require:

- 1) A .docx formatted version of the manuscript text (including legends for main figures, EV figures and tables). Please make sure that the changes are highlighted to be clearly visible.
- 2) Individual production quality figure files as .eps, .tif, .jpg (one file per figure). For guidance, download the 'Figure Guide PDF': (<https://www.embopress.org/page/journal/17574684/authorguide#figureformat>).
- 3) A .docx formatted letter INCLUDING the reviewers' reports and your detailed point-by-point responses to their comments. As part of the EMBO Press transparent editorial process, the point-by-point response is part of the Review Process File (RPF), which will be published alongside your paper.
- 4) A complete author checklist, which you can download from our author guidelines (<https://www.embopress.org/page/journal/17574684/authorguide#submissionofrevisions>). Please insert information in the checklist that is also reflected in the manuscript. The completed author checklist will also be part of the RPF.
- 5) Please note that all corresponding authors are required to supply an ORCID ID for their name upon submission of a revised manuscript.
- 6) It is mandatory to include a 'Data Availability' section after the Materials and Methods. Before submitting your revision, primary datasets produced in this study need to be deposited in an appropriate public database, and the accession numbers and

database listed under 'Data Availability'. Please remember to provide a reviewer password if the datasets are not yet public (see <https://www.embopress.org/page/journal/17574684/authorguide#dataavailability>).

12) Author contributions: You will be asked to provide CRediT (Contributor Role Taxonomy) terms in the submission system. These replace a narrative author contribution section in the manuscript.

13) A Conflict of Interest statement should be provided in the main text.

14) Every published paper now includes a 'Synopsis' to further enhance discoverability. Synopses are displayed on the journal webpage and are freely accessible to all readers. They include a short stand first (maximum of 300 characters, including space) as well as 2-5 one-sentences bullet points that summarizes the paper. Please write the bullet points to summarize the key NEW findings. They should be designed to be complementary to the abstract - i.e. not repeat the same text. We encourage inclusion of key acronyms and quantitative information (maximum of 30 words / bullet point). Please use the passive voice. Please attach these in a separate file or send them by email, we will incorporate them accordingly.

15) Include a Reagents and Tools Table as part of the Methods section, which can be downloaded from our author guidelines (<https://www.embopress.org/page/journal/17574684/authorguide#structuredmethods>)

***** Reviewer's comments *****

Referee #1 (Comments on Novelty/Model System for Author):

ORopuche virus infections suffer from a lack of diagnostic tools, resulting in many cases being misdiagnosed as dengue or chikungunya cases because of the similarity of the symptoms. The tools reported in this manuscript can have a strong impact to follow this type of outbreaks,

Referee #1 (Remarks for Author):

In this manuscript by Merchant et al describe the production of two recombinant antigens of Oropouche virus (OROV), the spike domain of the surface glycoprotein Gc (GcS) and the nucleocapsid protein (NP). They show that both recombinant antigens are recognized by the sera of mice experimentally infected with OROV but not from uninfected mice. Moreover, they use the antigens to immunize mice and show that the sera of GcS and GcS+NP immunized animals can neutralize OROV in cell culture. They further immunize llama and obtain single-chain antibody fragments (nanobodies) against both antigens. They characterize several of the nanobodies, which they show can be used as specific diagnostic tools. For this purpose, they use dimerized nanobodies by fusion with the Fc fragment of human immunoglobulin 1. They validate the potential of the biotinylated dimerized nanobodies, in particular one directed against NP, using samples collected from patients with acute viral illness that had been confirmed positive for OROV infection by PCR. Importantly, the same NP-directed nanobody also detected the antigen in samples from the recent OROV AM0088 strain outbreak in Brazil in 2022-2023. They also show that the nanobodies against Gc target conformational epitopes and can be used to monitor the location of the glycoproteins in infected cells, pointing to sites where it accumulates before budding of new particles. Moreover, the dimerized nanobodies against GcS neutralized in vitro infection by OROV of the corresponding strain with an IC50 of in the low nM range, and one of them kept the same neutralization potency against infections with the AM0088 strain currently circulating in Brazil.

Given the scarcity of tools to combat this neglected virus, the results provided in this manuscript are highly valuable and will accelerate ways of detecting this often mis-diagnosed pathogen. The manuscript is very clearly written, and the experiments were carefully designed.

I have only one comment concerning the characterization of the nanobodies, which ones target overlapping or non-overlapping epitopes. The authors found two classes of non-competing nanobodies targeting GcS and no competing nanobodies directed against NP. Only in the discussion they explain why: because NP will coat cellular RNAs, the antigen is in a form of a ribonucleoprotein (RNP) complex with multiple NP molecules, and immobilizing the RNP with any of the NP-specific nanobodies would leave NPs exposed and accessible for additional soluble nanobody (even the same one used for immobilization) to bind to the same epitope (this should have been raised in the results section already, without leaving the reader wondering until the discussion). The way to do the experiment would be to saturate the RNPs with soluble nanobody, and verify that it cannot bind to the sensor on which the same saturating nanobody was immobilized, but could bind to sensors coated with a non-competing nanobody.

Referee #2 (Remarks for Author):

Review

The paper describes a panel of Variable Heavy-chain domains of camelid Heavy-chain antibodies specific for the nucleoprotein and surface glycoprotein of OROV.

Thorough assessment demonstrates their efficacies in detecting OROV antibodies in vitro and in vivo.

The methods are well described results are appropriately presented and the paper is acceptable for publication after dealing with some minor comments (see below).

Results

correct Fig. references as shown below:

Purified antigens were used as capture antigens (Fig. 1B) in an indirect ELISA and incubated with serum from mice experimentally infected with OROV. Both N and Gc spike are recognised by IgG (Fig. 1C) and IgM (Fig. S1A)

Fig 1 D please add a definition of the % to the legend.

Fig 2 B. Please detail the number of points which were scored with n=2 and n=3

Discussion

Your paragraph:

Having established proof-of-principle that antigen detection can be used for OROV molecular diagnosis, it should be possible to develop a nanobody-based lateral flow device for diagnosing OROV infection at primary points-of-care. Rapid point-of-care diagnosis of OROV would enhance surveillance efforts, which could be especially important for pregnant women in endemic areas, and it would dramatically accelerate enrolment of patients into trials aimed at determining post-infection sequelae or the efficacy of future therapies.

Comment

Although generally true I wouldn't bargain on your current Ab system achieving the same sensitivity on a paper strip as in a homogenous assay, especially since you already observed sensitivity issues. Please reformulate in a way to make clear that only after you have solved the sensitivity issue one can think about lateral flow concepts

Your paragraph:

We have not investigated the cross-reactivity of our nanobodies with other Simbu serogroup N proteins.

Comment

Which others do you think would be necessary to test? In the face of M-segment reassortants such as IQTV (isolated from a febrile patient) and MDDV (isolated from a monkey) isn't it a good outcome that your anti-N-Abs provide the best results?

We thank the referees for their supportive statements and suggestions. We have addressed all their concerns, as detailed below.

Referee #1 (Comments on Novelty/Model System for Author):

ORopuche virus infections suffer from a lack of diagnostic tools, resulting in many cases being misdiagnosed as dengue or chikungunya cases because of the similarity of the symptoms. The tools reported in this manuscript can have a strong impact to follow this type of outbreaks,

Referee #1 (Remarks for Author):

In this manuscript by Merchant et al describe the production of two recombinant antigens of Oropouche virus (OROV), the spike domain of the surface glycoprotein Gc (GcS) and the nucleocapsid protein (NP). They show that both recombinant antigens are recognized by the sera of mice experimentally infected with OROV but not from uninfected mice. Moreover, they use the antigens to immunize mice and show that the sera of GcS and GcS+NP immunized animals can neutralize OROV in cell culture. They further immunize llama and obtain single-chain antibody fragments (nanobodies) against both antigens. They characterize several of the nanobodies, which they show can be used as specific diagnostic tools. For this purpose, they use dimerized nanobodies by fusion with the Fc fragment of human immunoglobulin 1. They validate the potential of the biotinylated dimerized nanobodies, in particular one directed against NP, using samples collected from patients with acute viral illness that had been confirmed positive for OROV infection by PCR. Importantly, the same NP-directed nanobody also detected the antigen in samples from the recent OROV AM0088 strain outbreak in Brazil in 2022-2023. They also show that the nanobodies against Gc target conformational epitopes and can be used to monitor the location of the glycoproteins in infected cells, pointing to sites where it accumulates before budding of new particles. Moreover, the dimerized nanobodies against GcS neutralized in vitro infection by OROV of the corresponding strain with an IC50 of in the low nM range, and one of them kept the same neutralization potency against infections with the AM0088 strain currently circulating in Brazil.

Given the scarcity of tools to combat this neglected virus, the results provided in this manuscript are highly valuable and will accelerate ways of detecting this often misdiagnosed pathogen. The manuscript is very clearly written, and the experiments were carefully designed.

I have only one comment concerning the characterization of the nanobodies, which ones target overlapping or non-overlapping epitopes. The authors found two classes of non-competing nanobodies targeting GcS and no competing nanobodies directed against NP. Only in the discussion they explain why: because NP will coat cellular RNAs, the antigen is in a form of a ribonucleoprotein (RNP) complex with multiple NP molecules, and immobilizing the RNP with any of the NP-specific nanobodies would leave NPs exposed and accessible for additional soluble nanobody (even the same one used for immobilization) to bind to the same epitope (this should have been raised in the results section already, without leaving the reader wondering until the discussion). The way to do the experiment would be to saturate the RNPs with soluble nanobody, and verify that it cannot bind to the sensor on which the same saturating nanobody was immobilized, but could bind to sensors coated with a non-competing nanobody.

We thank the reviewer for this excellent suggestion. We have performed the suggested experiment using biolayer interferometry, loading streptavidin biosensors with biotinylated nanobodies and then incubating them with N protein alone or in the presence of saturating

(25-fold molar) excess of competitor nanobody. These analyses, presented in Figures 2C and EV2E, confirm that the nanobodies against OROV N fall into three competition groups (NpA2, NpE2 and NpE3; NpF2 and NpG1; and NpD2). We have also updated the results text for the nanobody sandwich ELISA (Figure 2B) to explain immediately our hypothesis as to why no competition was seen for the N nanobodies.

Referee #2 (Remarks for Author):

Review

The paper describes a panel of Variable Heavy-chain domains of camelid Heavy-chain antibodies specific for the nucleoprotein and surface glycoprotein of OROV.

Thorough assesemnt demonstartes ther efficacies in detecting OROV antibodies in vitro and in vivo.

The methods are well described results are appropriately presented and thepaper is acceptable for publication after dealing with some minor comments (see below).

Results

correct Fig. references a shown below:

Purified antigens were used as capture antigens (Fig. 1B) in an indirect ELISA and incubated with serum from mice experimentally infected with OROV. Both N and Gc spike are recognised by IgG (Fig. 1C) and IgM (Fig. S1A)

Thank you for pointing out this error – the figure callouts have been updated as suggested.

Fig1 D please add a definition of the % to the legend.

We have defined the percentage infection more clearly in the legend, as follows:

“Percentage infection was calculated as number of plaques observed for each sample, divided by the mean number observed for the mock-immunised mice.”

Fig 2 B. Please detail the number of points which were scored with n=2 and n=3

We apologise for the lack of clarity. We have updated the legend to clarify that for the non-competing nanobodies the experiment was performed twice independently. One replicate was performed with a single fixed concentration of antigen and the second replicate used concentration gradient of antigen – to avoid complicating the paper we have not plotted this data, but it is included in the Source Data file that is deposited with the manuscript. We also updated the legend to confirm that, for the competing Gc spike nanobodies where we saw reciprocal competition between capture and detection nanobodies, the experiment was performed once. Updated legend text is as follows:

“For non-competing nanobodies, data are representative of 2 independent experiments. For competing Gc spike nanobodies the experiment was performed once.”

Discussion

Your pargarph:

Having established proof-of-principle that antigen detection can be used for OROV molecular diagnosis, it should be possible to develop a nanobody-based lateral flow device for diagnosing OROV infection at primary points-of-care. Rapid point-of-care diagnosis of OROV would enhance surveillance efforts, which could be especially important for pregnant

women in endemic areas, and it would dramatically accelerate enrolment of patients into trials aimed at determining post-infection sequelae or the efficacy of future therapies.

Comment

Althoug generally true I wouldn't bargain on your current Ab system achieving the same sensitivity on a paper strip as in a homogenous assay, especially since you already observe sensitivity issues. Please reformulate in a way to make clear that only after you have solved the sensitivity issue one can think about lateral flow concepts

We agree with the reviewer that overcoming sensitivity issues will be a key challenge for the future development of lateral flow devices – we apologise that we did not make this clear enough in the original text. We have updated the discussion of future lateral flow devices (Discussion paragraph 2) to highlight these challenges:

“Translation from an ELISA to lateral flow format may further reduce the sensitivity of antigen detection, although this could potentially be ameliorated by use of different pairs of capture and detection nanobody-Fc fusions and/or further protein engineering to enhance avidity, as mentioned above. Assuming that adequate sensitivity can be achieved then rapid point-of-care diagnosis...”

Your paragraph:

We have not investigated the cross-reactivity of our nanobodies with other Simbu serogroup N proteins.

Comment

Which others do you think would be necessary to test ? In the face of M-segment reassortants such as IQTV (isolated from a febrile patient) and MDDV (isolated from a monkey) isn't it a good outcome that your anti-N-Abs provide the best results?

The reviewer is 100% correct that the ability to detect infection by other, related orthobunyaviruses is an advantage and not a deficiency – we thank them for emphasising this. We have clarified our statement about cross-reactivity, as follows:

“We have not investigated the cross-reactivity of our nanobodies with other Simbu serogroup N proteins. Of most relevance at the other Simbu serogroup viruses that were identified in humans, OROV strains Iquitos and Madre de Dios [49,50], or non-human primates, OROV strain Perdões that was identified in black-tufted marmosets [51] and Manzanilla virus first identified in the red howler monkey [52]. N proteins from OROV strains BeAn19991, Iquitos, Madre de Dios and Perdões share 100% amino acid identity with OROV BeAn19991, thus all would be recognised by the nanobodies presented here. In a clinical context, definitive assignment of the OROV strain infecting an individual would require an orthogonal molecular technique like RT-PCR directed at the M segment (56.6–63.3% nucleotide identity across Oropouche virus strains) and/or full virus genome sequencing [13]. Manzanilla virus N [49] shares only 71.2% amino acid identity with OROV N, making cross-reactivity possible but much less likely. Identification of which (if any) OROV N nanobodies detect Manzanilla virus N awaits future experimental confirmation.”

Additional updates to the manuscript

Figure 1D was updated to show individual data points as per journal style.

Figure 3B has been updated because the values had accidentally not been background subtracted

Figure 3C has been updated to explicitly note which clinical samples were assayed only once

Figure 4C has been updated to show individual data points as per journal style

22nd Jul 2025

Dear Dr. Graham,

Thank you for the submission of your revised manuscript to EMBO Molecular Medicine. I am pleased to inform you that we will be able to accept your manuscript pending the following final amendments:

1) In the main manuscript file, please do the following:

- Correct order of manuscript sections: Abstract / Keywords / The Paper Explained / Introduction / Results / Discussion / Methods / Data Availability / Acknowledgements / Disclosure and Competing Interests Statement / References / Main Figure Legends / Tables / Expanded View Figure Legends.
- Please add the heading "Abstract" to the abstract.
- Add up to 5 keywords.
- In Methods, provide the statement that informed consent was obtained from all human subjects and confirm that the experiments conformed to the principles set out in the WMA Declaration of Helsinki and the Department of Health and Human Services Belmont Report.
- Rename "Conflict of interest" to "Disclosure and competing interests statement". We updated our journal's competing interests policy in January 2022 and request authors to consider both actual and perceived competing interests. Please review the policy <https://www.embopress.org/competing-interests> and update your competing interests if necessary.
- In data availability statement replace current text with "This study includes no data deposited in external repositories."
- Remove Table EV1 and upload it as a separate file.
- Please correct the reference citation in the text and reference list. In the text a reference should be cited by author and year of publication. Include a space between a word and the opening parenthesis of the reference that follows. In the reference list, citations should be listed in alphabetical order. Where there are more than 10 authors on a paper, 10 will be listed, followed by "et al.". Also, please remove DOIs. DOIs should only be used for preprints and datasets that have not been published. Please check "Author Guidelines" for more information.

<https://www.embopress.org/page/journal/17574684/authorguide#referencesformat>

2) Funding: Please make sure that information about all sources of funding are complete in both our submission system and in the manuscript. International Science Partnerships Fund (ISPF) Institutional Support Grant (ODA) from Research England is missing in our submission system.

3) Source data: Please provide source data for Figure EV2B, C and D.

4) The Paper Explained: Please add it to main manuscript file.

5) Synopsis:

- Synopsis image: Please simplify the provided visual abstract e.g., reduce number of graphs/blots.
- Please check your synopsis text and image before submission with your revised manuscript. Please be aware that in the proof stage minor corrections only are allowed (e.g., typos).

6) As part of the EMBO Publications transparent editorial process initiative (see our Editorial at <http://embomolmed.embopress.org/content/2/9/329>), EMBO Molecular Medicine will publish online a Review Process File (RPF) to accompany accepted manuscripts. This file will be published in conjunction with your paper and will include the anonymous referee reports, your point-by-point response and all pertinent correspondence relating to the manuscript. Let us know whether you agree with the publication of the RPF and as here, if you want to remove or not any figures from it prior to publication. Please note that the Authors checklist will be published at the end of the RPF.

7) Please provide a point-by-point letter INCLUDING my comments as well as the reviewer's reports and your detailed responses (as Word file).

I look forward to reading a new revised version of your manuscript as soon as possible.

Yours sincerely,

Zeljko Durdevic

Zeljko Durdevic
Senior Editor
EMBO Molecular Medicine

*** Instructions to submit your revised manuscript ***

- 1) a .docx formatted version of the manuscript text (including Figure legends and tables)
 - 2) Separate figure files*
 - 3) supplemental information as Expanded View and/or Appendix. Please carefully check the authors guidelines for formatting Expanded view and Appendix figures and tables at <https://www.embopress.org/page/journal/17574684/authorguide#expandedview>
 - 4) a letter INCLUDING the reviewer's reports and your detailed responses to their comments (as Word file).
 - 5) The paper explained: EMBO Molecular Medicine articles are accompanied by a summary of the articles to emphasize the major findings in the paper and their medical implications for the non-specialist reader. Please provide a draft summary of your article highlighting
 - the medical issue you are addressing,
 - the results obtained and
 - their clinical impact.This may be edited to ensure that readers understand the significance and context of the research. Please refer to any of our published articles for an example.
 - 6) Author contributions: the contribution of every author must be detailed in a separate section.
 - 7) EMBO Molecular Medicine now requires a complete author checklist (<https://www.embopress.org/page/journal/17574684/authorguide>) to be submitted with all revised manuscripts. Please use the checklist as guideline for the sort of information we need WITHIN the manuscript. The checklist should only be filled with page numbers where the information can be found. This is particularly important for animal reporting, antibody dilutions (missing) and exact values and n that should be indicated instead of a range.
 - 8) Every published paper now includes a 'Synopsis' to further enhance discoverability. Synopses are displayed on the journal webpage and are freely accessible to all readers. They include a short stand first (maximum of 300 characters, including space) as well as 2-5 one sentence bullet points that summarise the paper. Please write the bullet points to summarise the key NEW findings. They should be designed to be complementary to the abstract - i.e. not repeat the same text. We encourage inclusion of key acronyms and quantitative information (maximum of 30 words / bullet point). Please use the passive voice. Please attach these in a separate file or send them by email, we will incorporate them accordingly.
- You are also welcome to suggest a striking image or visual abstract to illustrate your article. If you do please provide a jpeg file 550 px-wide x 300-600px high.
- 9) A Conflict of Interest statement should be provided in the main text
 - 10) Please note that we now mandate that all corresponding authors list an ORCID digital identifier. This takes <90 seconds to complete. We encourage all authors to supply an ORCID identifier, which will be linked to their name for unambiguous name identification.

Currently, our records indicate that the ORCID for your account is 0000-0003-4547-4034.

Please click the link below to modify this ORCID:
Link Not Available

11) Include a Reagents and Tools Table as part of the Methods section, which can be downloaded from our author guidelines (<https://www.embopress.org/page/journal/17574684/authorguide#structuredmethods>)

Photos 400-800 DPI

*Additional important information regarding figures and illustrations can be found at <https://bit.ly/EMBOPressFigurePreparationGuideline>. See also figure legend preparation guidelines: <https://www.embopress.org/page/journal/17574684/authorguide#figureformat>

***** Reviewer's comments *****

Referee #1 (Comments on Novelty/Model System for Author):

The manuscript describe nanobodies capable of diagnosing Oropouche virus infection, a neglected virus that is currently causing major outbreaks in Brazil, and for which very little tools are available for detection.

Referee #1 (Remarks for Author):

the authors have adequately revised the manuscript, addressing my comments and also those of the other reviewer. I don't have any other request.

Referee #2 (Comments on Novelty/Model System for Author):

Use of adequate methodology results in novel set of specific antibodies for detection of OROV-Ags. Medical impact medium as OROV currently occurs only in South and Meso-Americas.

Referee #2 (Remarks for Author):

The manuscript is acceptable for publication

Referee #1 (Comments on Novelty/Model System for Author):

The manuscript describe nanobodies capable of diagnosing Oropouche virus infection, a neglected virus that is currently causing major outbreaks in Brazil, and for which very little tools are available for detection.

Referee #1 (Remarks for Author):

the authors have adequately revised the manuscript, addressing my comments and also those of the other reviewer. I don't have any other request.

No concerns are raised. We thank the reviewer for their time assessing our manuscript.

Referee #2 (Comments on Novelty/Model System for Author):

Use of adequate methodology results in novel set of specific antibodies for detection of OROV-Ags.

Medical impact medium as OROV currently occurs only in South and Meso-Americas.

Referee #2 (Remarks for Author):

The manuscript is acceptable for publication

No concerns are raised. We thank the reviewer for their time assessing our manuscript.

Additional updates to the manuscript

We have corrected a typographical error in the methods, confirming that we used 0.005% hydrogen peroxide (approx. 1.6 mM) in the TMB ELISA detection reagent (not 0.05% as previously written).

25th Jul 2025

Dear Dr. Graham,

We are pleased to inform you that your manuscript is accepted for publication and is now being sent to our publisher to be included in the next available issue of EMBO Molecular Medicine.
